# Understanding Intensity–Duration–Frequency (IDF) Curves Using IMERG Sub-Hourly Precipitation against Dense Gauge Networks

Alcely Lau and Ali Behrangi *

Department of Hydrology and Atmospheric Sciences, University of Arizona, Tucson, AZ 85721, USA
* Correspondence: behrangi@arizona.edu

**Abstract:** The design storm derived from intensity–duration–frequency (IDF) curves is the main input for hydrologic analysis or hydraulic design for flood control. The regions with higher flood risks due to extreme precipitation are often deficient in precipitation gauges. This study presents a detailed evaluation of IDF curves derived using IMERG Final half-hourly precipitation (V06), fitted with the widely used CDFs: Gumbel and MLE, Gumbel and MM, Pearson 3, and GEV. As benchmarks and following the same method, we also derived IDF curves using areal average gridded precipitation constructed from two dense gauges networks over (1) the WegenerNET Feldbach region in the Alpine forelands of Austria and (2) the gauge network of the Walnut Gulch Experimental Watershed, in a semiarid region of the United States. In both regions, the frequency analysis for return periods between 2 and 100 years was based on half-hourly rainfall and compared at a grid-scale with a spatial resolution of IMERG, $0.1° \times 0.1°$ lat/lon. The impact of order in which the gridded gauge-based precipitation average is performed within an IMERG grid was evaluated by computing two different Annual Maximum Series (AMS). In one, the average was computed before obtaining the AMS (AB-AMS), and in the other, the average was computed after obtaining the AMS for each gauge grid (AA-AMS) within the IMERG grid. The evaluation revealed that IMERG AMS agrees better with AB-AMS than AA-AMS for the two study regions. Lastly, it was found that the use of Gumbel distribution in calculating IMERG IDF curves results in better agreement with the ground truth than the use of the other three distributions studied here. The outcomes should provide valuable knowledge for the application of IMERG precipitation over regions with sparse gauges.

**Keywords:** IDF curves; remote sensing; precipitation; IMERG; WegenerNET; Walnut Gulch; annual maximum series; probability distribution function; gridded precipitation; areal average





## 1. Introduction

Observed climate change, including increases in frequency and intensity of extreme precipitation, has caused impacts on key infrastructure systems, resulting in economic losses, disruptions of services, and impacts on well-being [1]. With high confidence, the IPCC Working Group II [1] reported that between 2010 and 2020, human mortality from floods, droughts, and storms was 15 times higher in highly vulnerable regions compared to regions with very low vulnerability. Moreover, infrastructure systems and basic services will be increasingly vulnerable if design standards do not account for changing climate conditions. The risks to life, economies, and infrastructures due to floods are listed with at least a medium confidence level for Europe, Asia, Central and South America [1].

The intensity–duration–frequency (IDF) curves are graphical representations that summarize conditional probabilities (frequencies) of rainfall depths or average intensities. Design storms derived from IDF curves are among the most common type of processed data for hydrologic and hydraulic design of urban drainage systems, evaluating the endurance of hydraulic structures, and assessing regional flood vulnerabilities [2,3]. Consequently,

updated and reliable IDF curves are crucial in the design and planning of urban and rural settlements and infrastructures, especially those that contribute to flood control.

Furthermore, the frequency analysis used to derive the IDF curves requires precipitation measurements at local, fine scales with long-term records. As a result, dense observational networks are necessary to adequately capture rainfall variability, particularly at fine temporal and spatial scales [4]. However, it is often not the case for many regions due to the high cost, data confidentiality, and time-consuming procedures involved in data acquisition and sharing systems [3]. Therefore, the construction of IDF curves for most countries around the world remains a major challenge [5].

Potentially, satellite data can provide a good alternative for developing IDF curves due to their quasi-global coverage and sub-daily temporal sampling. Several studies have focused on the assessment of satellite-based precipitation products with weather station measurements. However, few studies have addressed the application of remote sensing products for the derivation of IDF curves [5]. Some of these studies proposed techniques to mix satellite-based with point-gauge-based precipitation. For example, Endreny and Imbeah [6] suggested a short-record IDF approach by mixing the Ghanaian Meteorological Service Department (GMSD) ground gauge data and Tropical Rainfall Measuring Mission (TRMM) GEV-II distribution parameters. Ombadi et al. [7] proposed a framework to develop IDF curves over the contiguous United States (CONUS), accumulating the Precipitation Estimation from Remotely Sensed Information Using Artificial Neural Networks–Climate Data Record (PERSIANN-CDR) in durations of 1–3 day IDFs and then evaluated them against the National Oceanic and Atmospheric Administration (NOAA) Atlas 14. Sun et al. [3] demonstrated an improvement in Singapore IDF curves by integrating the Global Satellite Mapping of Precipitation (GSMaP), providing sub-daily rainfall, with the Bartlett–Lewis rectangular pulses (BLRP) model, to disaggregate the daily in situ rainfall to hourly rainfall. Furthermore, some studies compared the performance of IDF curves using remote-sensing-based precipitation against gridded gauge-based data. Marra et al. [8] compared the use of rainfall estimates from a ground-based C-band weather radar and from a high-resolution satellite precipitation product, CMORPH (HRC), and its gauge-adjusted version (CHRC), for the derivation of sub-daily IDF curves covering different climates over the eastern Mediterranean. Noor et al. [5] evaluated the ability of the following remote sensing products to generate sub-daily IDF curves in Peninsular Malaysia: GSMaP, GSMaP Near Real-Time (GSMaP_NRT), PERSIANN, and TRMM. Kyaw et al. [9] developed IDF curves for Yangon, Myanmar, using four satellite precipitation datasets, namely GPM IMERG, TRMM, GSMaP_NRT, and GSMaP_GC. They reported that IDF curves generated using satellite rainfall products also showed less bias in IMERG IDF curves than those obtained for other products [9].

As a whole, these studies imply that IDF curves developed from satellite-based rainfall data tend to deviate from curves developed from the point gauges data. Apart from that, Noor et al. [5] recommended that the CDFs should be estimated for both observed and remote sensing data to allow a better comparison with remote sensing rainfall products. Moreover, all the studies recognized that due to the improvement in satellite-based precipitation products, it is essential to evaluate the sub-hourly performance of other high-resolution satellite-based rainfall products, such as NASA Global Precipitation Measurement (GPM) Integrated Multi-satellitE Retrievals for GPM (IMERG).

In view of the previous studies, the aim of this work is to advance knowledge on the application of satellite-based sub-hourly precipitation data for the development of IDF curves. Specifically, the main focus lies on analyzing the ability of IMERG Final half-hourly precipitation to derive IDF curves against two spatially dense gauge networks: the WGEW network in a semiarid region of Arizona, United States, and the WegenerNET FBR in the southeastern Alpine forelands of Austria. The goal is to investigate how we can construct the IDF curves from IMERG (short for IMERG Final), so it can yield the closest results to that obtained from the gauge networks. The hope is that the methods and details discussed in this work can guide the generation of IDF curves from satellites when and where we

do not have dense gauge networks. Therefore, two separate regions were selected to assess the consistency of the outcomes. To obtain IDF curves, as discussed in Section 2.3, we calculate the precipitation Annual Maximum Series (AMS) for each dataset for event durations between 0.5 and 24 h. We also analyze the effect produced by the order of operation for gauges: averaging gridded gauge-based precipitation in each IMERG grid before calculating the AMS (AB-AMS) or calculating AMS at each gauge grid and after that averaging the AMSs (AA-AMS). Furthermore, we perform a frequency analysis fitting the AMSs to the four most widely used CDFs for this kind of analysis: Gumbel and MLE, Gumbel and MM, Pearson 3, and GEV. For the frequency analysis, we consider the return periods 2, 5, 10, 20, 30, 50, and 100 years. Then, we derive the IDF equations at a grid level considering the AA-AMS and AB-AMS cases (areal average order) in each region. Finally, we compare IMERG AMS and IDF curves against those obtained with the averaged gauge-based using different performance metrics. In brief, the following research questions were discussed in this study:

- How different are the IMERG AMS and the gauge-based AMS?
- How does areal averaging before the AMS (AB-AMS) and after the AMS (AA-AMS) affect the precipitation AMS?
- Which CDF fits the IMERG AMS better?
- What is the performance of IMERG IDF curves at grid and local scales?

## 2. Materials and Methods

### 2.1. Data

The study focused on the derivation of IDF curves using the half-hour precipitation estimate from IMERG. As the ground measurement benchmark to assess the performance of IMERG, the study uses two operational long-term weather observation networks at the 1–10 km scale: the Walnut Gulch Experimental Watershed (WGEW) and the WegenerNet Feldbach Region (FBR) [10].

#### 2.1.1. Integrated Multi-SatellitE Retrievals for GPM (IMERG)

GPM mission is an international constellation of satellites that provide global multi-satellite observations of rain and snow. Where, IMERG is the multi-satellite precipitation algorithm that combines the precipitation measurements from the GPM constellation sensors. The latest version of IMERG-Final product (V06) [11] is used in this study, with data obtained from 2001 to 2020. IMERG provides gridded precipitation data with half-hour temporal and $0.1° \times 0.1°$ spatial resolution by combining precipitation estimates from two main satellite sources: (a) passive microwave observations from the GPM core and GPM constellation satellites, and (b) geostationary infrared images and through the PERSIANN-CCS precipitation retrieval method. The Climate Prediction Center morphing Kalman filter (CMORPH-KF) Langrangian approach uses the PMW and IR estimates to create half-hourly estimates. IMERG has three products, "Early", "Late", and "Final". The algorithm runs twice in near-real time to produce the Early product ~4 h after observation time (AOT) and the Late product ~14 h AOT. In comparison, the Final product has a latency of ~3.5 months and is created through bias-adjusting the estimates to match monthly gauge analysis. The Early and Late products are the baseline for calibrating the Final product with gauge climatological coefficients [12].

Figure 1 shows the IMERG grids within the study areas. The study in the Feldbach region (FBR), Austria, focuses on two IMERG grids with centered coordinates of: (a) 15.85° east and 46.95° north (FBR 1), and (b) 15.95° east and 46.95° north (FBR 2). We used the half-hour IMERG-Final precipitation data from 1 January 2007 to 31 December 2020. The study in the Walnut Gulch Experimental Watershed, Arizona, focuses on two IMERG grids with centered coordinates of: (a) −110.05° east and 31.75° north (WGEW 1), and (b) −109.95° east and 31.75° north (WGEW 2). Here, we used the half-hour IMERG-Final precipitation data from 1 January 2001 to 31 December 2020. The IMERG Final L3 Half Hourly (V06) product can be accessed from https://disc.gsfc.nasa.gov/ (accessed on 30 November 2021).

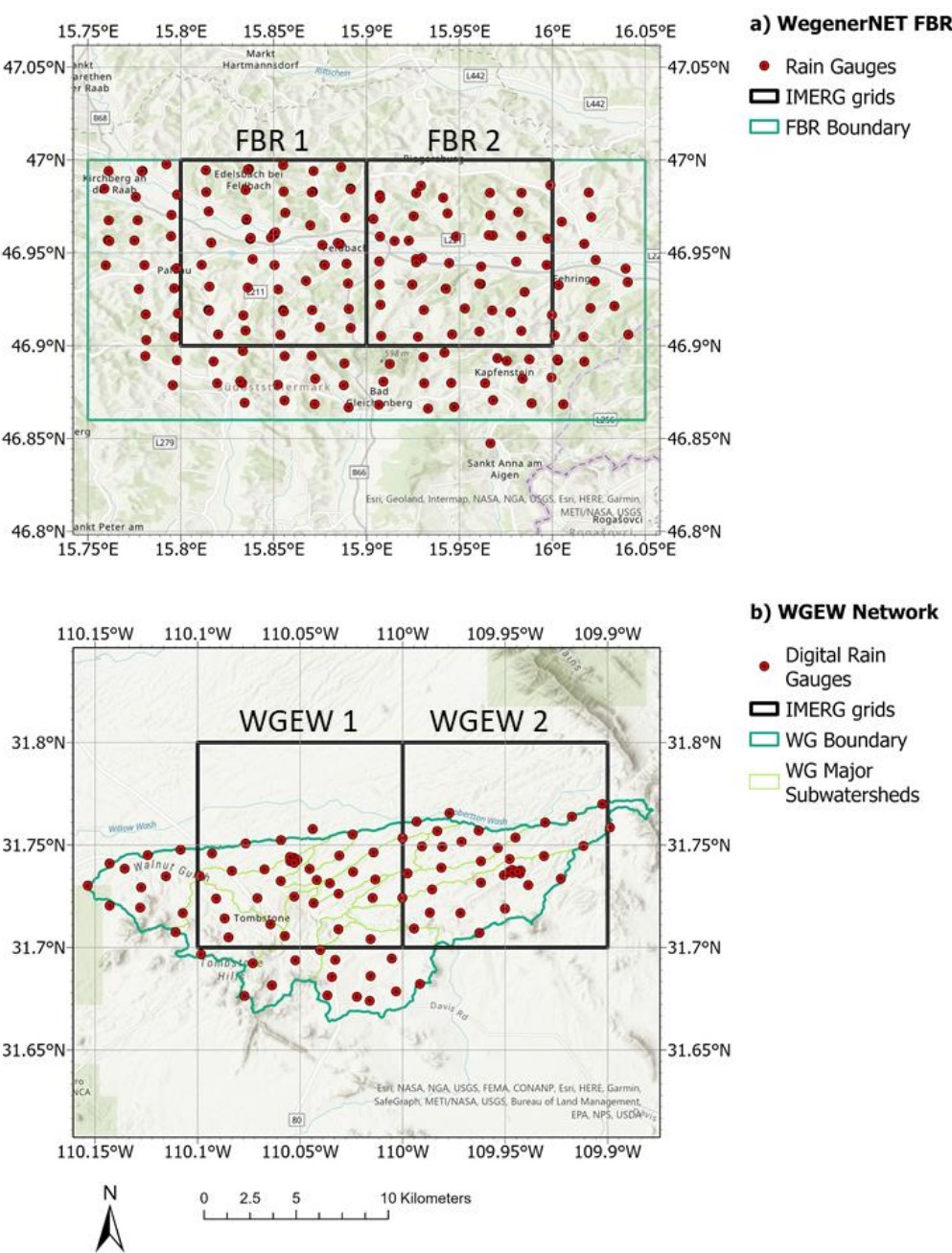

**Figure 1.** Maps of the study areas: (**a**) WegenerNet Feldbach Region (FBR) network, Austria, and (**b**) Walnut Gulch Experimental Watershed (WGEW), Arizona. The black boxes are the IMERG grids.

2.1.2. WegenerNet Feldbach Region (FBR) Network

The WegenerNet Feldbach Region (FBR) network is long-term hydrometeorological station network operated by the Wegener Center of the University of Graz since 1 January 2007 [13].

The FBR network is located between 15.76° and 16.04° north and 46.87° and 47.00° east in the county of Feldbach, southeast of Styria state, Austria. The FBR is part of the southeastern Alpine forelands, with the Raab River dividing the region. Thus, the terrain elevation of FBR ranges from 250 to 500 m above MSL. Moreover, its diverse orography generates seasonal and local climate variabilities. The main weather systems over the FBR region are small-scale convective thunderstorms that usually occur during the summer rainfall. Additionally, hailstorms, warm-and-cold front rainfall, Genoa lows, and occasional strong winter storms occur [13,14].

Figure 1a shows the locations of meteorological stations within the FBR. The FBR network is composed of 155 meteorological stations with an average interstation distance of about 1.4 km, covering a region of about 22 km (from west to east) × 16 km (from north to south), with an effective area of around 300 km$^2$. Therefore, the FBR network has an average density of about 0.5 stations per square kilometer. Furthermore, WegenerNET FBR interpolated using the inverse-distance weighted from the neighbor stations method to create a 200 × 200 m gridded precipitation data set based on its 155 tipping bucket gauges [10,13]. Thus, we used the gridded half-hour FBR total precipitation data from 1 January 2007 to 31 December 2020. Data for the FBR can be accessed at https://wegenernet.org/portal/ (accessed on 30 November 2021).

2.1.3. Walnut Gulch Experimental Watershed (WGEW) Network

The U.S. Department of Agriculture, Agricultural Research Service, Southwest Watershed Research Center (USDA-ARS-SWRC) operates and maintains the Walnut Gulch Experimental Watershed (WGEW) network. The WGEW is a long-term precipitation gauge network established in August 1953.

WGEW is located between 37.66° and 31.77° north and −110.16° and −109.87° east in a semiarid region of the southeast of Tucson, Arizona. The watershed elevation ranges from 1220 to 1950 m above MSL with approximately 149 km$^2$ of drainage area. Furthermore, the precipitation over this region is influenced by the following weather systems: ~60% due to airmass convection thunderstorms during the summer (monsoon) season, ~35% winter frontal systems with rare hail or snow, and ~5% due to tropical depressions [15,16].

Figure 1b shows the locations of rain gauges within the WGEW. Until 1999, the WGEW operated with an analog weighing–recording rain gauge network. From 1 January 2000 to the present, the WGEW measurements mainly came from a digital weighing–recording network of rain gauges. Currently, the WGEW network is composed of 88 rain gauges with an average density of about 0.6 rain gauges per square kilometer. In detail, the rain gauges, designed by SWRC, measure the accumulated rainfall depths at 1 min intervals during periods of rainfall (minimum detectable precipitation is 0.25 mm) [16].

Moreover, there is a 100 × 100 m gridded precipitation data set that is the result of the multiquadric-biharmonic (MQB) spatial interpolation. Garcia et al. [17] found that the MQB interpolation method generally outperforms the IDW interpolation method applied to convective and stratiform events over WGEW. Here, we used the gridded half-hour WGEW total precipitation data measured with the digital rain gauges from 1 January 2001 to 31 December 2020. Data related to the WGEW can be accessed at https://www.tucson.ars.ag.gov/dap/ (accessed on 20 October 2021). Table 1 provides a summary about the precipitation datasets applied in this study.

**Table 1.** Precipitation datasets used in this study.

| Gridded Rainfall | Spatial Resolution | Temporal Resolution | Period | Reference |
|---|---|---|---|---|
| **IMERG Final** | 0.1° × 0.1° | Half-hour | Corresponding to the ground networks period | Huffman et al. [12] |
| **FBR Network** | 200 × 200 m | Half-hour | 2007 to 2020 | Kirchengast et al. [13] |
| **WGEW Network** | 100 × 100 m | Half-hour | 2001 to 2020 | Garcia et al. and Goodrich et al. [16,17] |

*2.2. Matching the IMERG Grid to Gauge Cells*

Considering the high density of the FBR and WGEW gauge networks, we will evaluate the IMERG AMS and IDF relationships at individual IMERG grids.

In the following sections, the term "cell" is used to differentiate the IMERG grid from the gauge network grids. Thus, the term cell refers to the area of each gridded-gauge

precipitation. For the study, we selected the IMERG grids with the highest coverages over the study areas. Eight IMERG grids cover the FBR. However, only two IMERG grids are inside the FBR boundaries providing 100% coverage over that area (Figure 1a). This means that an IMERG grid with 0.1° × 0.1° spatial resolution is equivalent to an area of the FBR gridded observations of 39 cells to the east by 59 cells to the north. Thus, for one value of IMERG precipitation there are 2301 FBR precipitation cells.

In WGEW, six IMERG grids cover the watershed, but none of the IMERG grids fit entirely inside the WGEW boundaries. Therefore, we used the two IMERG grids that provide coverage of ~93% over WGEW (Figure 1b). An IMERG grid with 0.1° × 0.1° spatial resolution is equal to an area of the WGEW gridded observations of 95 × 95 cells. A ~93% coverage corresponds to an area of the WGEW gridded observations of 95 cells to the east by 89 cells to the north. Hence, for one grid of IMERG we used 8455 WGEW cells.

### 2.3. Obtain the Annual Maximum Series (AMS)

Figure 2 summarizes the methodology followed for the development and evaluation of the IDF curves. Regardless of the data source, the AMSs were obtained by adding up the half-hour precipitation to construct durations of 0.5, 1, 1.5, 2, 2.5, 3, 6, and 24 h. The accumulating process for each year and for each duration started on 1 January at 00:00 UTC and it stopped on 31 December at 23:30 UTC. Then, the maximum annual accumulated precipitation was computed for the eight duration series.

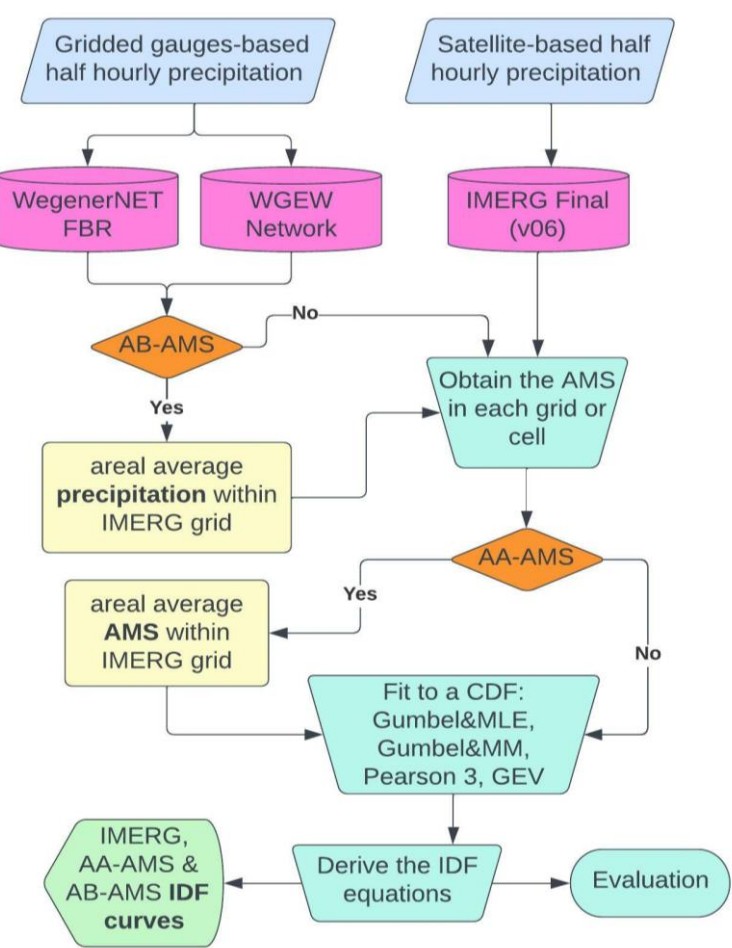

**Figure 2.** Study workflow.

For the IMERG half-hour precipitation data, the AMS was obtained for the grids FBR 1, FBR 2, WGEW 1, and WGEW 2. Then, we created the following two study cases. The first case was the areal average after obtaining the AMS (AA-AMS). The gauge-based AMS

was obtained for each individual cell of the gauge gridded precipitation (cell) and then the AMS areal average was computed for all the cells inside each IMERG grid. The second case was the areal average before obtaining the AMS (AB-AMS). In the AB-AMS, we used the original half-hour precipitation of the cells inside each IMERG grid, and then the AMS was obtained over the area of the IMERG grid.

### 2.4. Compute the Intensity

The intensity of precipitation is defined as the precipitation depth divided by the duration associated with the accumulated precipitation:

$$i = \frac{p}{d} \tag{1}$$

where $i$ is the intensity of precipitation (usually expressed in mm/hr), $p$ is the precipitation depth in units of length, and $d$ is the duration of the precipitation event in units of time.

The analysis of the annual maxima assumes that the largest event of precipitation in one year is independent of the largest event in any other year. However, there is a possibility that the second largest event in one year is greater than the largest event in another year, especially in shorter durations, i.e., the analysis may not provide the true maximum amounts for a particular duration. For that reason, we followed the WMO [18] recommendation to multiply the precipitation intensity by an adjustment factor for a fixed observation frequency in the order of 1.13 (for one half-hour observation) to 1.00 (for more than 24 half-hour observations).

### 2.5. Frequency Analysis: Fitting a PDF to the AMS

The return period of an annual maximum precipitation event is associated with a probability that the magnitude of this event is equaled or exceeded once during every T years [2]. The relationship between the exceedance probability and the return period is described in the following equation:

$$P = \frac{1}{T} \tag{2}$$

where $P$ is the exceedance probability and $T$ is the return period in years. Thus, for a precipitation event with a return period of 5 years there is a probability of 20% that an event with the same or higher magnitude and duration may repeat within a period of 5 years.

Hence, a probability density function (PDF) generates a relationship between the intensity of precipitation, the duration, and a return period. With interest in obtaining the related probability of each extreme event, we applied four theoretical cumulative distribution functions (CDF) commonly used in hydrology for the analysis of extreme events. Table 2 shows the PDF or CDF formulas for Gumbel or Extreme Value Type I with the estimation of parameters using Maximum Likelihood (Gumbel and MLE), Gumbel with the estimation of parameters using the Method of Moments (Gumbel and MM), Pearson type III (Pearson 3), and General Extreme Value (GEV).

**Table 2.** The PDF or CDF formulas used for the frequency analysis.

| Method | PDF or CDF [19] |
|:---:|:---:|
| **Gumbel** | $F(x) = \exp\left[-\exp\left(-\frac{x-\xi}{\alpha}\right)\right]$ |
| **Pearson 3** | $f(x) = \lvert\beta\rvert[\beta(x-\xi)]^{\alpha-1}\frac{\exp[-\beta(x-\xi)]}{\Gamma(\alpha)}$ |
| **GEV** | $F(x) = \exp\left\{-\left[1 - \frac{k(x-\xi)}{\alpha}\right]^{1/k}\right\}$ |

In this study, $F(x)$ is the exceedance probability function and $x$ is the rainfall intensity associated to a selected return period and duration. $\xi$, $\alpha$, $\beta$, and $k$ are parameters of the theoretical distribution function. $\Gamma(\ )$ is the gamma function [19].

Fitting a distribution to the data can lead to different results depending on the data source and the type of the CDF. Therefore, the Kolmogorov–Smirnoff test (KS-test) was performed to compare the goodness of fit between the CDFs [20,21]. We consider a 90% confidence and a two-tailed test to calculate the KS test statistic measures.

### 2.6. Regression Model: Derive the IDF Equation

Subsequently, the series of intensity, duration, and return period were fitted to a curve. In hydrology, the equation that represents the relationship between the intensity of precipitation, duration, and frequency has the general form [22]:

$$i(d,\ T) = \frac{C}{(d^v + t_0)^n} \tag{3}$$

where $i(d, T)$ is the intensity of precipitation for a particular return period ($T$), usually expressed in mm/hr, and $d$ is the duration of the precipitation event in units of time. The $C$, $t_0$, $v$, and $n$ are parameters to be determined according to the series of each return period.

The IDF relationships depend on location and geographical conditions. In consequence, the literature does not present a consensus about the best empirical model. However, some local IDF studies over Panama [23] and Vietnam [24] stated that Talbot's equation outperforms Kimijima and Bernard. Lau and Perez (2015) found that Talbot's equation has a better performance for short durations (i.e., <2 h) and comparable results after ~2 h than Bernard's equation [23].

For the reasons above, we consider the most general equation and widely used empirical formula of Talbot, which from Equation (3), $v = 1$ and $n = 0$.

$$i = \frac{a}{d + b} \tag{4}$$

where $i(d, T)$ is the intensity of precipitation for a particular return period ($T$), in mm/hr. The $d$ is the duration of the precipitation event in hours. The $a$ and $b$ are the regression parameters for each return period ($T$), determined by the least square method.

### 2.7. Metrics for Evaluation

Furthermore, we used the derived IDF equations for each CDF and each return period to compute the precipitation intensity (mm/hr) associated with durations between 0.5 and 24 h with increments of 0.5 h. Its purpose was to use those values for the evaluation of the derived IMERG AMSs and IMERG IDF curves from a grid and local (cell) scale perspective.

#### 2.7.1. Grid-Scale Performance

The grid-scale evaluation was approached using the following method. First, we compared the IMERG AMS against the AA-AMS and AB-AMS. For the AMS comparison, we considered the following statistic metrics from Table 3: the inter-annual maximum mean, the mean bias ratio (ratio), the Pearson correlation, and the precipitation differences (bias) for each duration using the FBR (14 years) and WGEW (20 years) precipitation records.

Then, we evaluated the grid-scale performance of the IMERG IDF curves using the corresponding gauge-based precipitation IDF curves as the benchmark. The evaluation considers the location (FBR and WGEW), type of the CDFs, order of the areal average during the process (AA-AMS and AB-AMS), event duration, and return period. For the IDF assessment, we computed the following statistic metrics: precipitation intensity differences (Bias), Nash–Sutcliffe efficiency coefficient (NSE) [25], and Kling–Gupta efficiency skill score (KGEss) [26,27].

**Table 3.** Statistical metrics used for the evaluation of the AMS and IDF equations. $x$ represents the IMERG values, $y$ the gauge-based values, $\mu$ is the mean, and $S$ is the standard deviation.

| Statistic | Formula | Range | Optimal Value |
|---|---|---|---|
| Mean bias ratio | $Ratio = \frac{\mu_x}{\mu_y}$ | 0 to 1 | 1 |
| Pearson correlation | $pcorr = \rho = \frac{1}{n-1} \sum_{i=1}^{n} \left( \frac{x_i - \mu_x}{S_x} \right) \left( \frac{y_i - \mu_y}{S_y} \right)$ | $-1$ to 1 | 1 |
| Bias | $Bias = x_i - y_i$ | $-\infty$ to $+\infty$ | 0 |
| Nash–Sutcliffe efficiency coefficient | $NSE = 1 - \frac{\frac{1}{n} \sum_{i=1}^{n} (x_i - y_i)^2}{\frac{1}{n-1} \sum_{i=1}^{n} (y_i - \mu_y)^2}$ | 0 to 1 | 1 |
| Kling–Gupta efficiency | $KGE = 1 - \sqrt{\left( 1 - \frac{S_x}{S_y} \right)^2 + \left( 1 - \frac{\mu_x}{\mu_y} \right)^2 + (1 - \rho)^2}$ | $-\infty$ to 1 | 1 |
| Kling–Gupta efficiency skill score | $KGEss = \frac{KGE + 0.4142}{\sqrt{2}}$ | $-\infty$ to 1 | 1 |
| Percent of relative error | $RE = \left( \frac{x_i - y_i}{y_i} \right) * 100\%$ | $-\infty$ to $+\infty$ | 0 |

### 2.7.2. Local-Scale Performance

Here, we compare the AMS and IDF of each cell within the IMERG grids against the IMERG AMSs and IDF curves. In particular, this analysis aims to assess the local performance of the satellite-retrieved precipitation IDF curves to represent the local variability of the precipitation over a grid area ($0.1° \times 0.1°$). In fact, this analysis could provide insights into the local-scale reliability of using IMERG to derive IDF curves over ungauged regions. Therefore, we computed the percent of relative error (RE) before deriving the IDF by comparing the IMERG AMS against each cell's AMS. Then, we computed the RE of the IMERG IDF against each cell's IDF inside each IMERG grid.

## 3. Results and Discussion

The evaluation process was performed on a grid-scale. However, considering that the results between the neighboring grids within the same region led to similar findings, we decided to present, in the following section, the plots for the grids FBR 2 and WGEW 1. The plots for FBR 1 and WGEW 2 are available in the Appendix A. Additionally, Tables 4 and 5 provide summaries of the regions because the values are the average of the two grids for each region.

### 3.1. Statistical Comparison between the IMERG AMS and the Gauge-Based AMS

In general, the performance of IMERG AMS varies with the study areas. Table 4 summarizes the ratio, bias, and Pearson correlation for FBR and WGEW. These metrics were obtained by comparing IMERG AMS against the gauge-based AMSs (AA-AMS and AB-AMS), according to the metrics in Table 3. Figures 3 and 4 show the annual maximum time series for FBR 2 and WGEW 1, respectively. The FBR time series has a record length of 14 years, from 2007 to 2020. In comparison, the WGEW time series record length is 20 years, from 2001 to 2020.

In FBR, IMERG tends to underestimate the AMS for precipitation events with durations shorter than 2 h, as can be seen from the FBR ratio and bias (in Table 4) and in Figure 3. In contrast, IMERG tends to overestimate the FBR gauge-based AMSs for precipitation events with durations longer than 2 h (Figure 3).

Figure 4 and Table 4 illustrate that IMERG tends to underestimate the WGEW gauge-based AMSs overall. Although Nashwan et al. [28] reported that IMERG overestimates in Egypt, a hot dessert climate, our results agree with Kyaw et al. [9] and Chen et al. [29]. This could be related to a few factors; among those are: (1) errors in passive microwave or infrared-based precipitation retrievals (or due to the sensors' large footprints) used as input in IMERG; (2) the process of averaging precipitation retrievals in time and space used in IMERG can change the distribution of the precipitation values, leading to an

increase in precipitating area and a decrease in the values of intense precipitation [30]; and (3) the evaporation of precipitation over dry and hot regions such as the WGEW and false alarms [9,29,31]. These factors may have different impacts in different regions. For example, it is expected that the underestimation of intense precipitation due to the averaging plays a more important role in the WGEW region (where convective systems are frequent in summer) than the FBR region. Furthermore, from [32] it can be concluded that the poorer performance of GPROF over WGEW might be due to the coarse identification of surface type at 1° in GPROF, in which the semiarid environment of WGEW is assigned to the "high vegetation" and "moderate vegetation" classes. Furthermore, it should be noted that the gauge-based adjustment employed in IMERG is at a monthly scale, so it may also negatively impact the distribution of precipitation at daily or shorter durations. Nevertheless, the bias between IMERG AMS and gauge-based AMSs decreases with longer durations for WGEW. It is likely that the biases in shorter durations tend to be smoothed by the accumulation of IMERG rainfall half-hourly into longer durations.

Furthermore, the Pearson correlation coefficients (pcorr) indicate that IMERG has difficulties in representing the temporal distribution of the rainfall AMS. However, it tends to improve at longer durations (e.g., 24 h). Although the FBR presents the lowest bias, the pcorr (Table 4) suggests that IMERG is not capturing the temporal patterns over the FBR very well. The pcorr over WGEW indicates that IMERG is following the temporal rainfall patterns over this region better than over the FBR region. This could be due to the fact that the detection of intense convective precipitation events, that are frequent over the WGEW region, are often easier for remote sensing products than the detection of non-convective precipitation events that are more frequent over the FBR than the WGEW region.

In FBR, the contradiction between low bias, good ratio, and poor pcoor values infers that the errors from different events inside the time series may cancel each other. For example, looking at the 14 years annual maximum time series of FBR (Figure 3), IMERG overestimated the maximum precipitation in 2014 almost in the same magnitude that it underestimated the maximum precipitation in 2020 at durations longer than 2.0 h.

**Table 4.** Ratio, bias, and Pearson correlation of IMERG AMS against gauge-based AMS for the precipitation event durations of 0.5, 1, 1.5, 2, 2.5, 3, 6, and 24 h.

| Region | Statistic | IMERG against | 0.5 h | 1 h | 1.5 h | 2 h | 2.5 h | 3 h | 6 h | 24 h |
|--------|-----------|---------------|-------|-----|-------|-----|-------|-----|-----|------|
| **FBR** | Ratio | AA-AMS | 0.63 | 0.74 | 0.86 | 0.97 | 1.03 | 1.07 | 1.14 | 1.10 |
| | | AB-AMS | 0.82 | 0.94 | 1.04 | 1.14 | 1.21 | 1.24 | 1.28 | 1.16 |
| | Bias | AA-AMS | −7.69 | −7.05 | −4.24 | −0.98 | 1.16 | 2.59 | 5.85 | 5.93 |
| | | AB-AMS | −2.80 | −1.24 | 1.00 | 3.91 | 5.94 | 7.23 | 10.16 | 8.90 |
| | Pcorr | AA-AMS | −0.18 | −0.11 | −0.11 | −0.13 | −0.12 | −0.08 | −0.19 | 0.02 |
| | | AB-AMS | −0.09 | −0.06 | −0.10 | −0.10 | −0.05 | 0.01 | −0.13 | 0.01 |
| **WGEW** | Ratio | AA-AMS | 0.27 | 0.35 | 0.45 | 0.53 | 0.59 | 0.63 | 0.75 | 0.80 |
| | | AB-AMS | 0.35 | 0.44 | 0.54 | 0.63 | 0.68 | 0.73 | 0.84 | 0.86 |
| | Bias | AA-AMS | −14.10 | −16.58 | −15.22 | −13.60 | −12.43 | −11.35 | −8.30 | −8.34 |
| | | AB-AMS | −9.74 | −11.73 | −10.72 | −9.13 | −8.28 | −7.40 | −5.02 | −5.65 |
| | Pcorr | AA-AMS | 0.45 | 0.39 | 0.35 | 0.35 | 0.38 | 0.39 | 0.38 | 0.48 |
| | | AB-AMS | 0.45 | 0.32 | 0.29 | 0.28 | 0.32 | 0.33 | 0.31 | 0.45 |

**Table 5.** Kolmogorov–Smirnoff test statistics (D) and *p* values according to the CDFs.

| Region | CDF | IMERG | | AA-AMS | | AB-AMS | |
|---|---|---|---|---|---|---|---|
| | | D | *p* Value | D | *p* Value | D | *p* Value |
| FBR | GEV | 0.14 | 0.89 | 0.16 | 0.81 | 0.14 | 0.88 |
| | Gumbel and MLE | 0.16 | 0.79 | 0.16 | 0.80 | 0.15 | 0.85 |
| | Gumbel and MM | 0.16 | 0.80 | 0.18 | 0.70 | 0.16 | 0.79 |
| | Pearson 3 | 0.18 | 0.69 | 0.19 | 0.66 | 0.19 | 0.65 |
| WGEW | GEV | 0.12 | 0.88 | 0.12 | 0.88 | 0.12 | 0.87 |
| | Gumbel and MLE | 0.14 | 0.80 | 0.12 | 0.87 | 0.13 | 0.81 |
| | Gumbel and MM | 0.14 | 0.74 | 0.13 | 0.83 | 0.13 | 0.81 |
| | Pearson 3 | 0.33 | 0.65 | 0.18 | 0.81 | 0.13 | 0.80 |

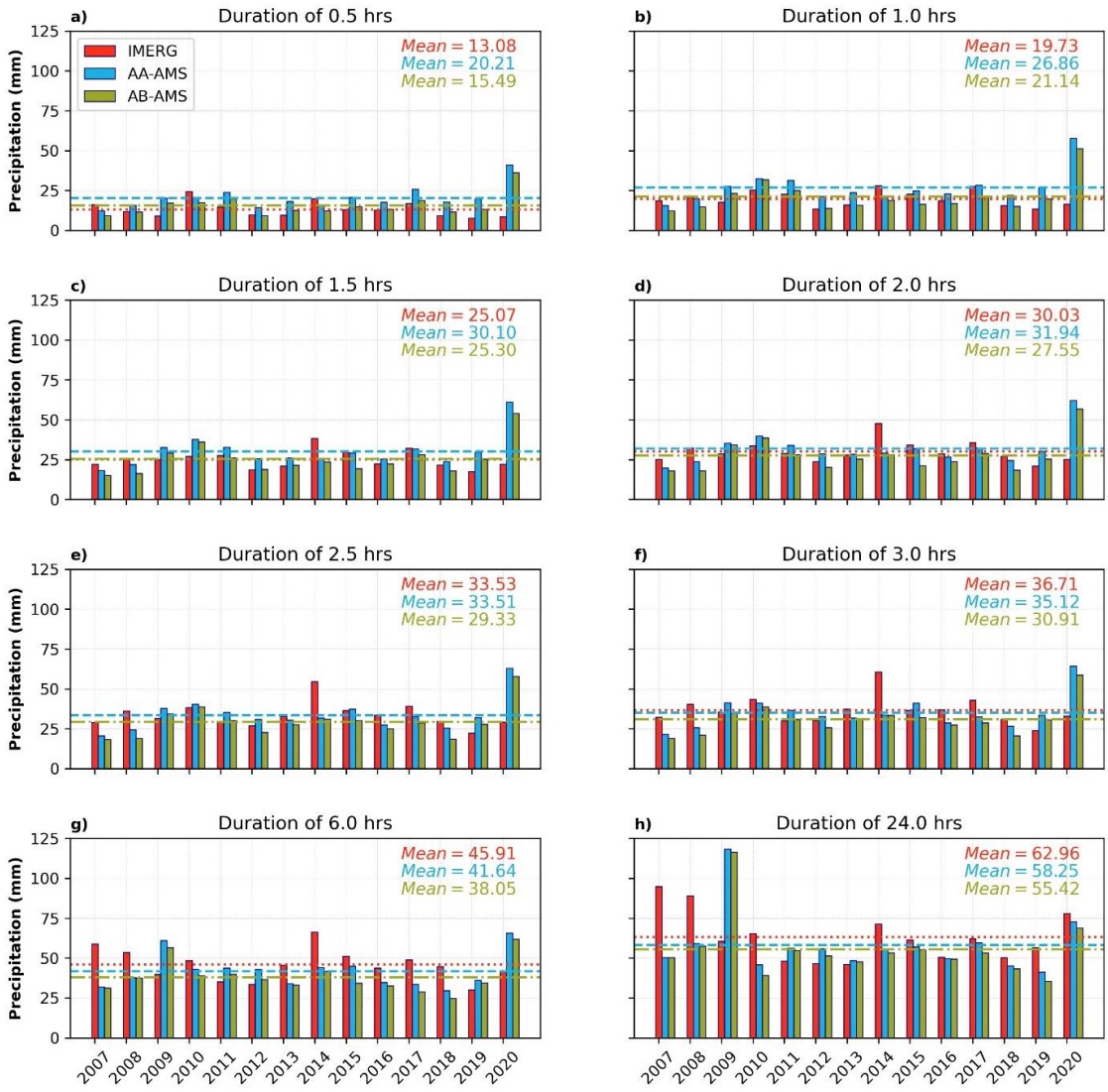

**Figure 3.** Annual maximum time series for FBR 2 by durations. The subfigures illustrate event durations of 0.5-h (**a**), 1-h (**b**), 1.5-h (**c**), 2-h (**d**), 2.5-h (**e**), 3-h (**f**), 6-h (**g**), 24-h (**h**). In all the subfigures, the red bars are the IMERG AMS, light blue bars are the AA-AMS, and the green bars are the AB-AMS. The dashed lines are the mean AMS for each case.

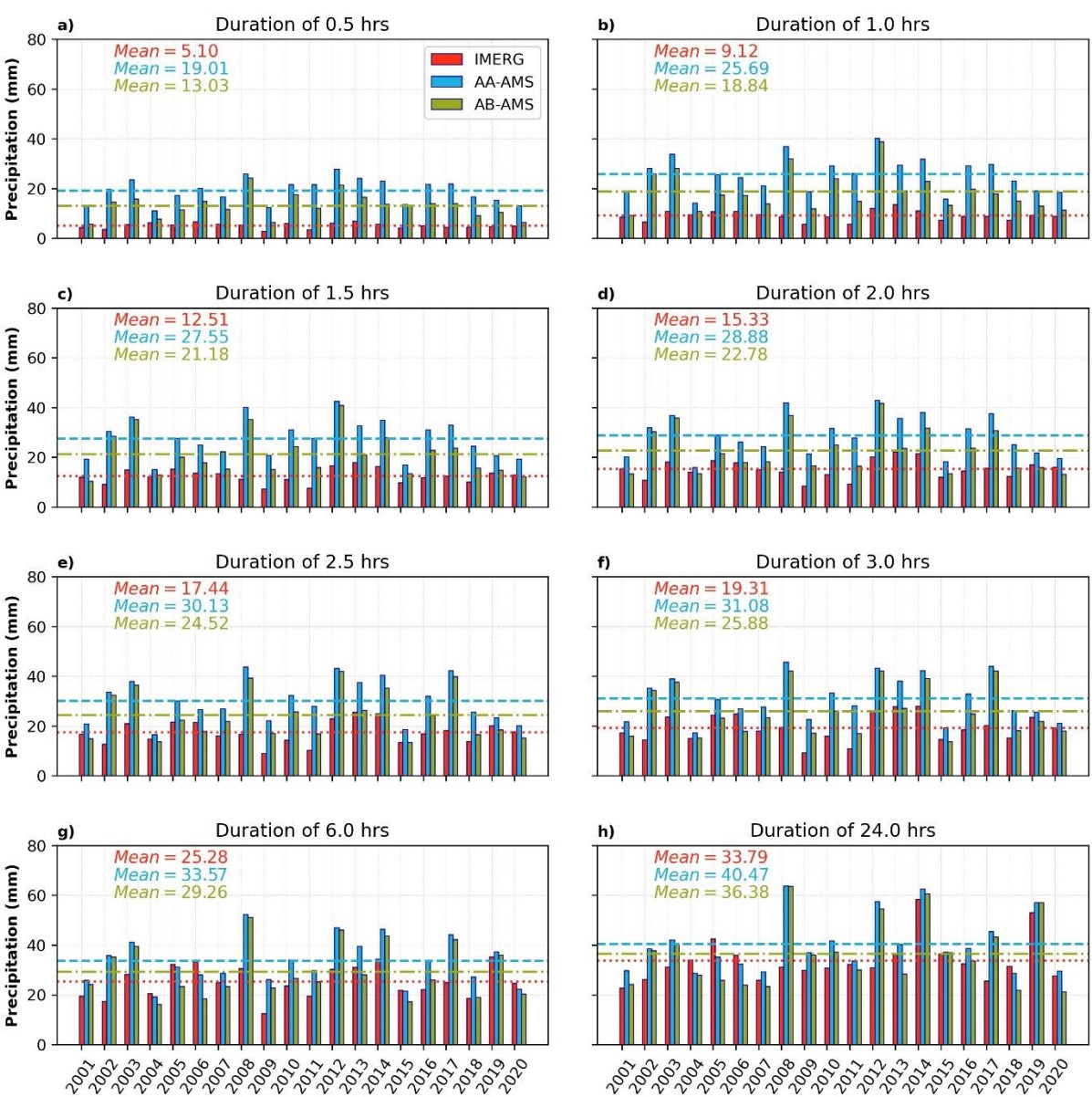

**Figure 4.** Annual maximum time series for WGEW 1 by durations. The subfigures illustrate event durations of 0.5-h (**a**), 1-h (**b**), 1.5-h (**c**), 2-h (**d**), 2.5-h (**e**), 3-h (**f**), 6-h (**g**), 24-h (**h**). In all the subfigures, the red bars are the IMERG AMS, light blue bars are the AA-AMS, and the green bars are the AB-AMS. The dashed lines are the mean AMS for each case.

*3.2. AA-AMS versus AB-AMS*

We visually compared the IMERG AMS against the gauge-based AMSs using quantile–quantile plots (qq-plots). The qq-plots facilitate the identification of shifts in location, shifts in scale, and outliers between the series. This comparison is better aligned with the purpose of the frequency analysis because it focuses on matching the rainfall amounts and does not identify if they come from the same rainfall event. Figure 5 illustrates the sorted values of the IMERG AMS against the sorted values of the AA-AMS (Figure 5a,d) and the AB-AMS (Figure 5b,e). Then, Figure 5c,f show the sorted values of the AA-AMS against the sorted values of the AB-AMS.

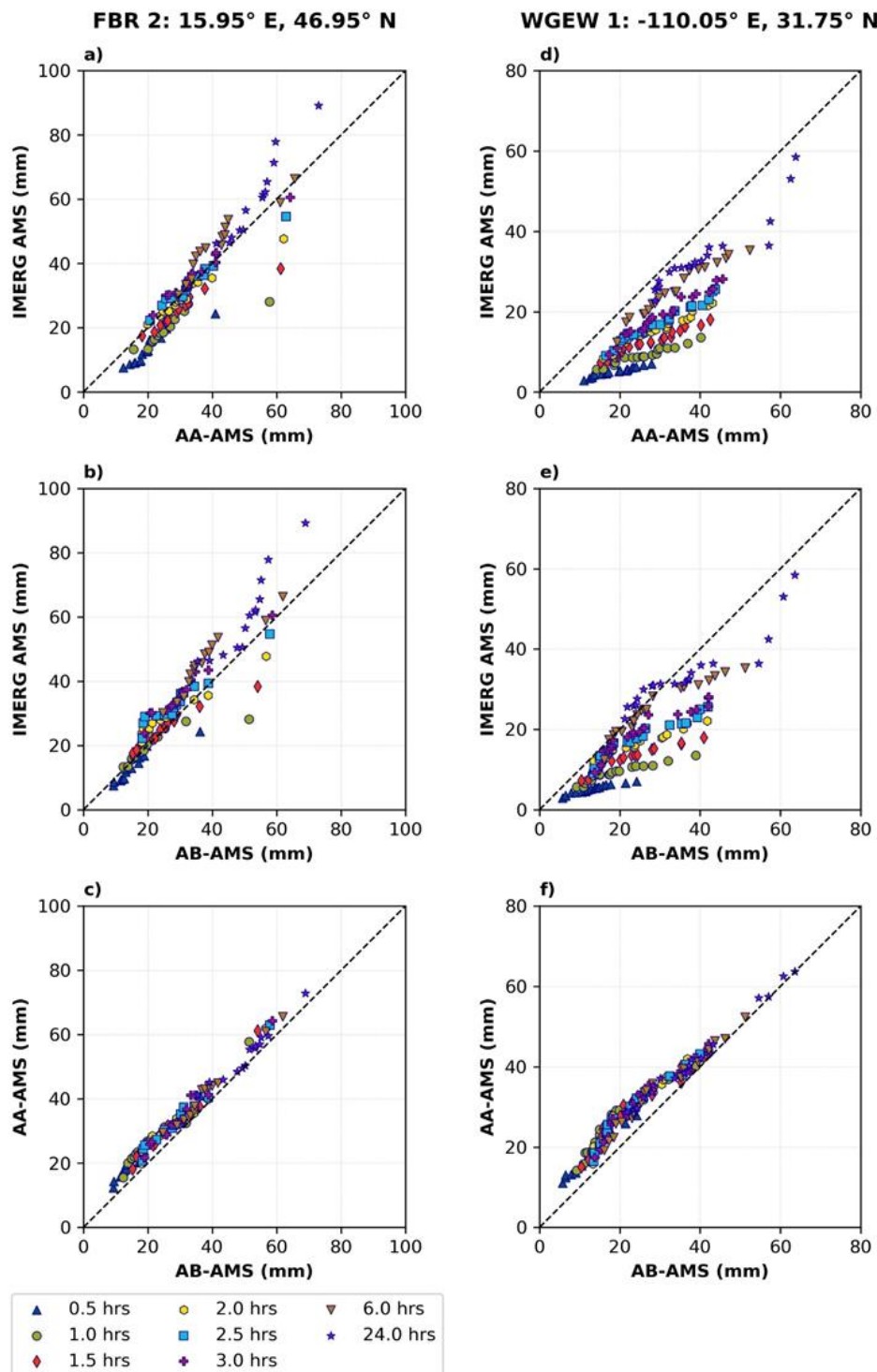

**Figure 5.** QQ-plots comparing the AMSs for FBR 2 (**left**) and WGEW 1 (**right**). From top to bottom: IMERG AMS vs. AA-AMS, IMERG AMS vs. AB-AMS, and AA-AMS vs. AB-AMS. The markers are the durations 0.5, 1, 1.5, 2, 2.5, 3, 6, and 24 h. The 45° dashed line is a reference line.

As previously mentioned, Figure 5 shows that IMERG estimates the annual maximum precipitation over the FBR better than the WGEW according to the historical gauge records. In the case of the FBR, the IMERG annual maximum rainfall is more similar to the amount obtained with the AA-AMS for duration between 2 and 3 h (Figure 5a). In contrast, the IMERG annual maximum rainfall is almost the same as the AB-AMS amounts for durations between 0.5 and 2 h (Figure 5b). Nonetheless, in both comparisons (Figure 5a,b), IMERG

overestimates the maximum rainfall in most of the years for 6 and 24 h. In WGEW, IMERG underestimates the maximum rainfall for most of the years in almost the whole range of the evaluated event durations. Clearly, IMERG estimates better if we compare it against AB-AMS (Figure 5e). This study does not aim to propose an adjustment factor to improve IMERG in regions with deficient gauges. However, the qq-plots for WGEW demonstrate that IMERG AMS has different shifts in scale for each duration that should be considered for adjustments.

Figure 5c,f demonstrate that AA-AMS leads to a higher amount of precipitation than AB-AMS, affecting the IDF curve calculations. On average, AA-AMS has about 18% higher values than AB-AMS for the same precipitation event. Coincidentally, this average percentage is similar for both regions.

### 3.3. CDF Goodness of Fit

Before fitting a CDF to the data, we examined the goodness of fitness of the four CDFs: Gumbel and MLE, Gumbel and MM, Pearson 3, and GEV and MLE with the IMERG, AA-AMS, and AB-AMS datasets of each duration and each grid using the Kolmogorov–Smirnoff goodness of fit test. Table 5 summarizes the KS-test average statistic (D) and the statistical significance (*p*-value).

All the datasets could acceptably fit any of the four CDFs (*p*-value > 0.05). Nonetheless, the KS-test indicates that GEV and MLE provides the best goodness of fit (Maximum *p*-value = 0.89) for all the datasets in both regions, which agrees with the results obtained by Noor et al. [5] applying negative log-likelihood goodness of fit tests and by Kyaw et al. [9]. These agreements may be related to the fact that the selection of the appropriate distribution is influenced by the record length. As Kyaw et al. recommended, these studies could be repeated in the future when observed and satellite data will be available for a longer period [9].

In second place is the Gumbel and MLE (maximum *p*-value = 0.87), and Pearson 3 obtained the worst goodness of fit (maximum *p*-value = 0.81) for all the datasets in both regions.

### 3.4. Grid-Scale: Performance of the IMERG IDF Equations

The following section covers the comparisons of the IMERG IDFs against the AA-AMS and AB-AMS IDFs using the different equations (but corresponding among the products) for deriving the IDFs. Figure 6 illustrates that the KGEss and the NSE values can vary considerably due to the type of the CDF and return period. Overall, NSE shows less sensitivity to changes in the return period than KGEss. Furthermore, this sensitivity is larger for shorter than longer return periods for both NSE and KGEss.

Overall, the KGEss and NSE imply that the IMERG IDF curves fitted with both of the Gumbel probability distributions outperform the other distributions. The lowest KGEss and NSE were obtained with the GEV and Pearson 3 for FBR and WGEW, respectively. Often, the scores decrease by increasing the return period. Hence, the highest scores are for the 2 year return period. The exception is Figure 6b, where the highest scores are between 5 and 10 year return periods.

Figure 6 shows that the comparison of the IMERG IDF curves against the AB-AMS IDF curves (Figure 6b,d) displays higher KGEss and NSE than comparing the IMERG IDF curves with AA-AMS IDF curves, and in all cases Gumbel and MLE show better scores. By using Gumbel and MLE, Figure 7 demonstrates that the precipitation intensities obtained with AB-AMS have better resemblance to IMERG IDF curves across all return periods considered.

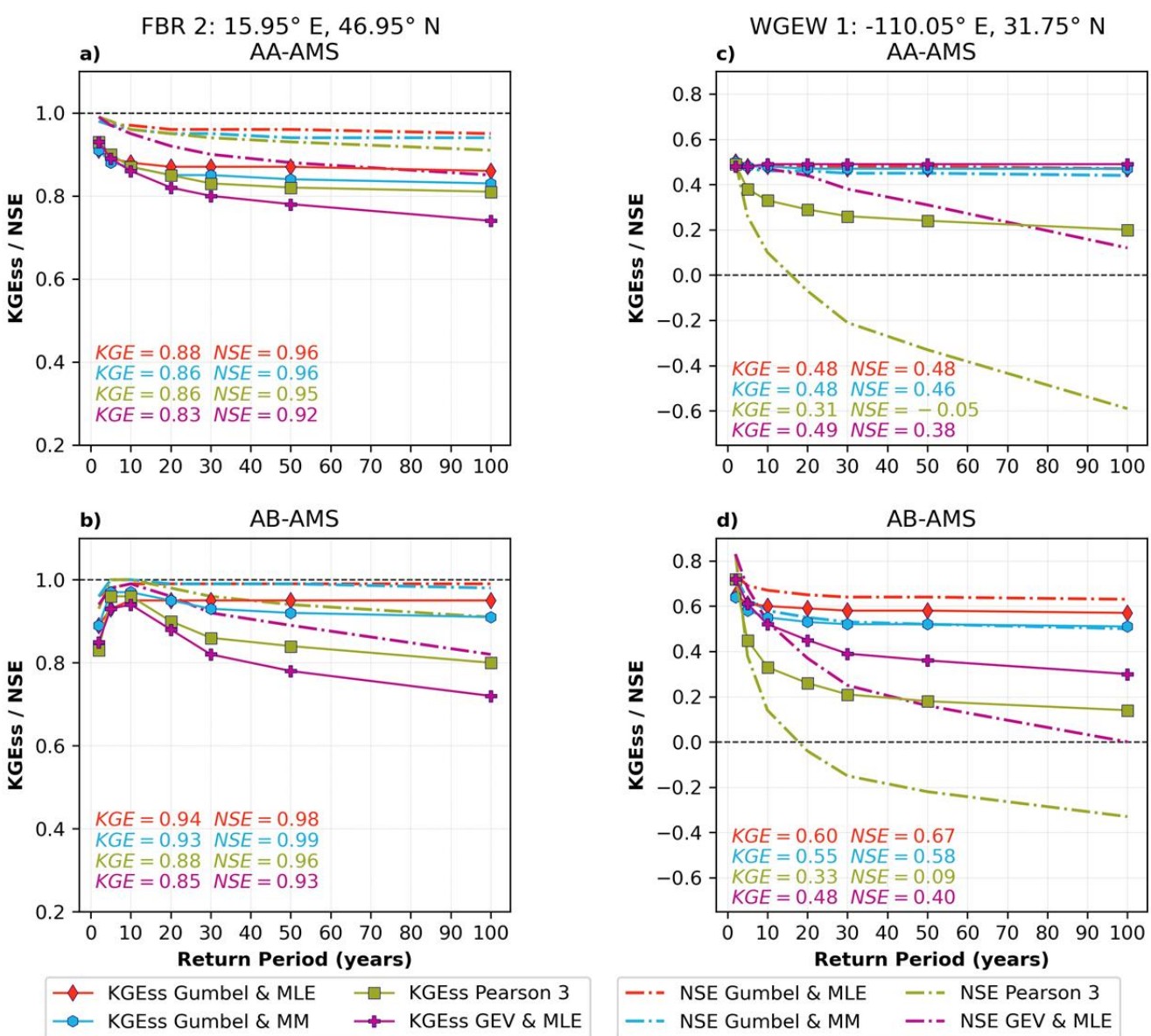

**Figure 6.** KGEss and NSE obtained for the four CDFs and the return periods 2, 5, 10, 20, 30, 50, and 100 years over FBR 2 (**left**) and WGEW 1 (**right**). IMERG IDF vs. AA-AMS IDF (subplots **a**,**c**) and IMERG IDF vs. AB-AMS IDF (subplots **b**,**d**). The solid lines with markers are for the KGEss and the dashed lines are for the NSE.

Figures 8 and 9 illustrate the precipitation intensity biases between IMERG IDFs and the gauge-based IDFs for 2, 5, 10, and 100 year return periods and for event durations between 0.5 and 12 h. It is evident that after 12 h the biases stay relatively unchanged. The negative values in Figures 8 and 9 mean that IMERG IDFs underestimate the precipitation intensities for the shorter durations. In the FBR, the biases vary between −25 and 4 mm/h and usually converge to zero after 5 h. In the case of the WGEW, the bias range is −65 to 0 mm/h and bias converges to zero after 7–9 h. Again, the comparison of IMERG against AB-AMS shows lower biases than AA-AMS in any grid. IMERG IDFs fitted with Gumbel and MLE tend to produce the lowest biases.



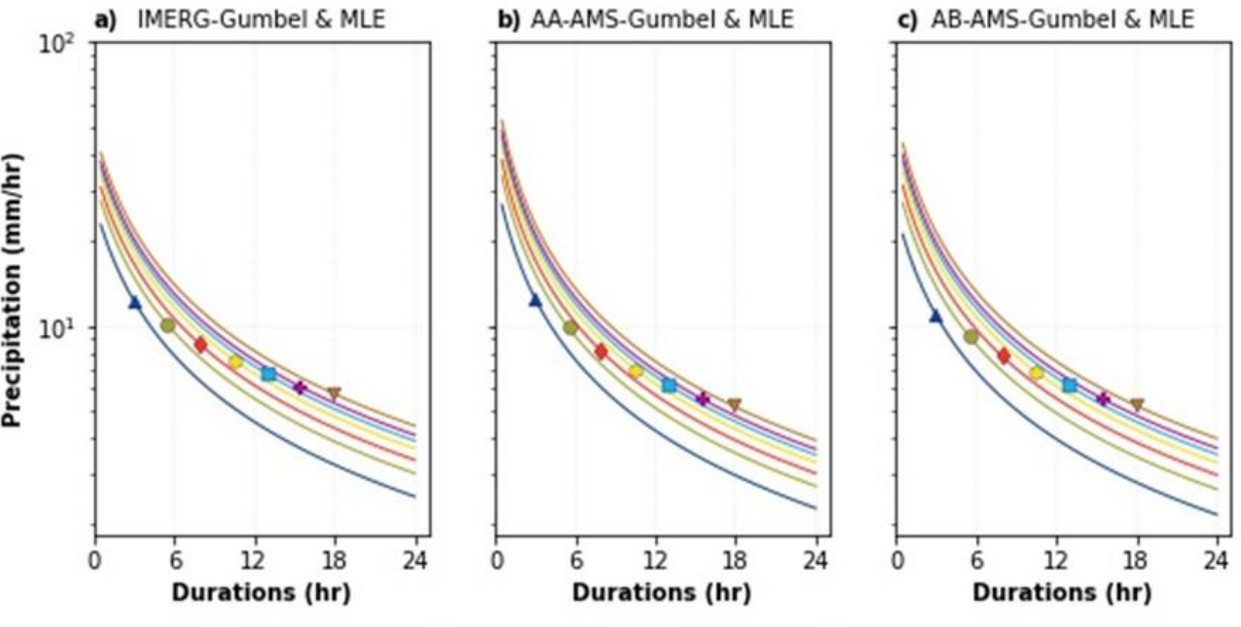

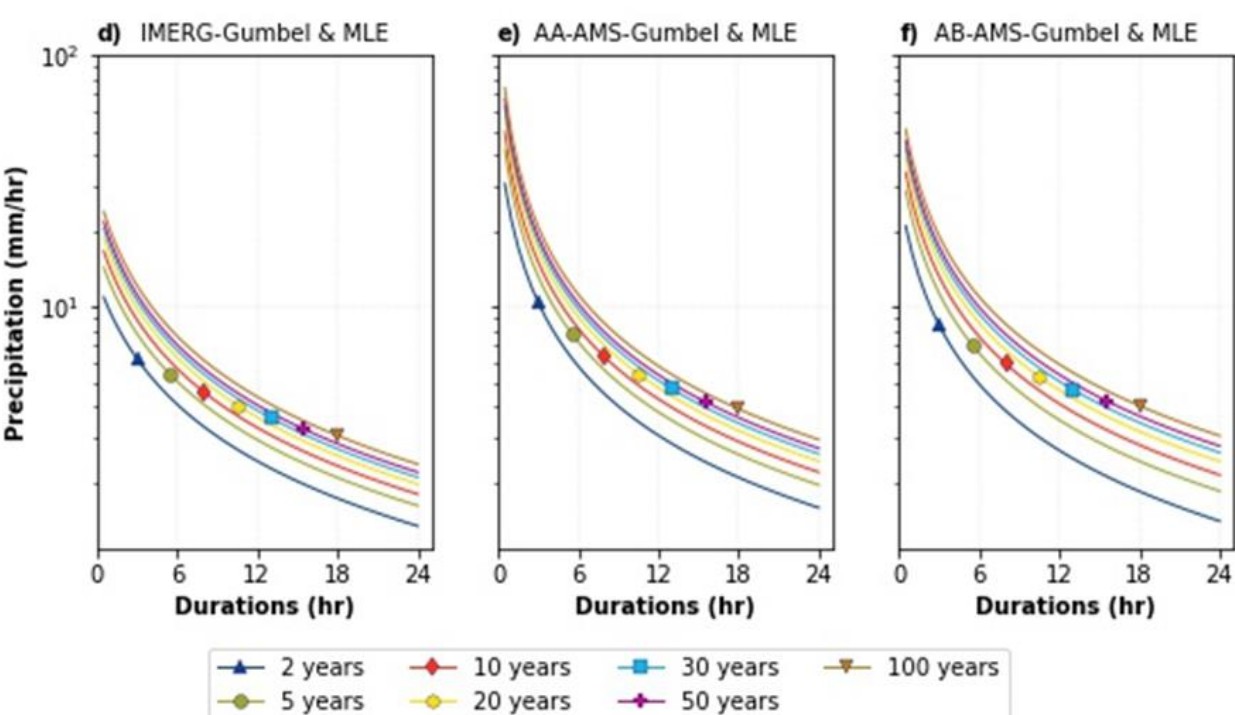

**Figure 7.** IDF curves fitted with Gumbel and MLE over FBR 2 (**top**) and WGEW 1 (**bottom**) for the 2, 5, 10, 20, 30, 50, and 100 year return periods and durations from half-hour to 24 h. From left to right: IMERG, AA-AMS, and AB-AMS. The marker's purpose is only to distinguish the IDF curves.

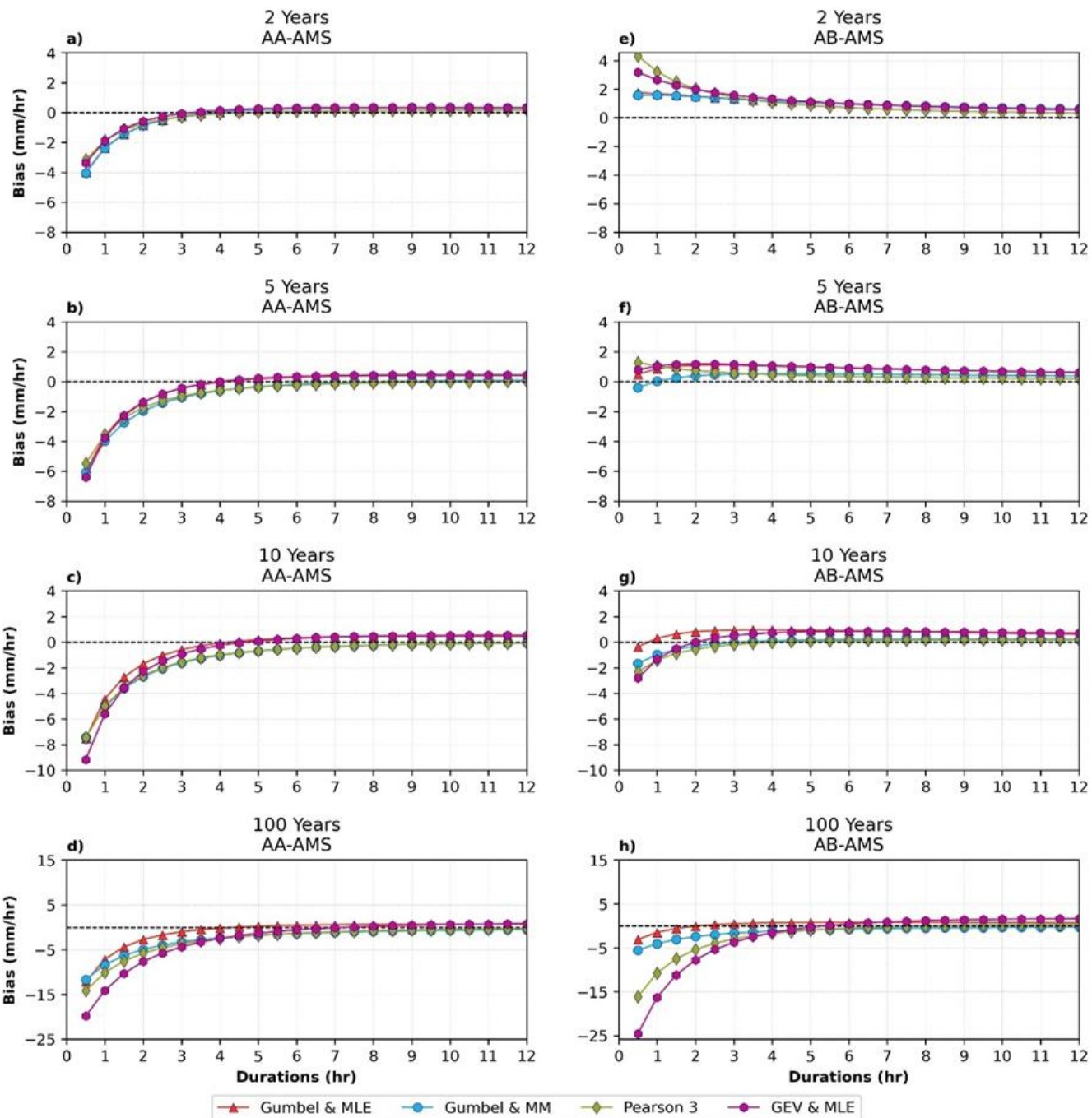

**Figure 8.** Precipitation intensity biases between IMERG IDFs and the gauge-based IDFs for 2, 5, 10, and 100 year return periods and event durations between 0.5 and 12 h over the FBR 2. IMERG IDF vs. AA-AMS IDF (**left**) and IMERG IDF vs. AB-AMS IDF (**right**).

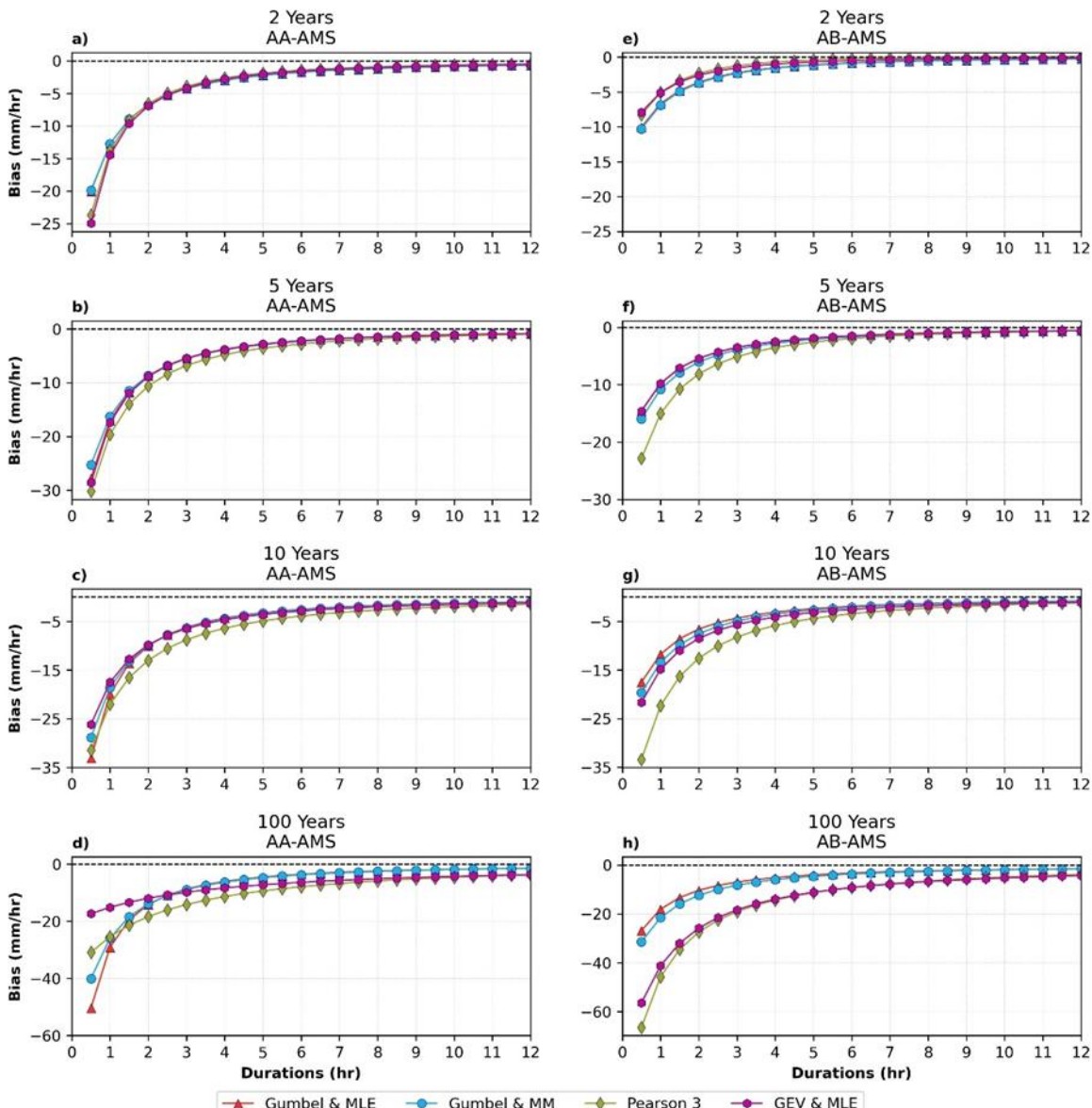

**Figure 9.** Precipitation intensity biases between IMERG IDFs and the gauge-based IDFs for 2, 5, 10, and 100 year return periods and event durations between 0.5 and 12 h over the WGEW 1. IMERG IDF vs. AA-AMS IDF (**left**) and IMERG IDF vs. AB-AMS IDF (**right**).

### 3.5. Assessment of IMERG AMSs and IDF Equations at Local Scale

Considering the potential application of IMERG precipitation for IDF development in an ungauged area or with a poor density of weather stations, we performed a comparison of the AMS and IDF between IMERG and each cell within the same grid.

Figure 10 summarizes the RE of each cell AMS against the IMERG AMS within the same grid. In this figure, we are presenting the results for the four studied grids. The box sizes indicate that overall precipitation spatial distribution tends to be more uniform over the WGEW than over the FBR. The boxplot's whiskers indicate that IMERG tends to show lower RE when it underestimates the precipitation. However, when IMERG overestimates the precipitation the RE is higher. Regarding the overestimation cases, it could be associated to occurrence of light rainfall that evaporates before it can reach the land surface, while from the satellite's perspective it is raining [31,32].

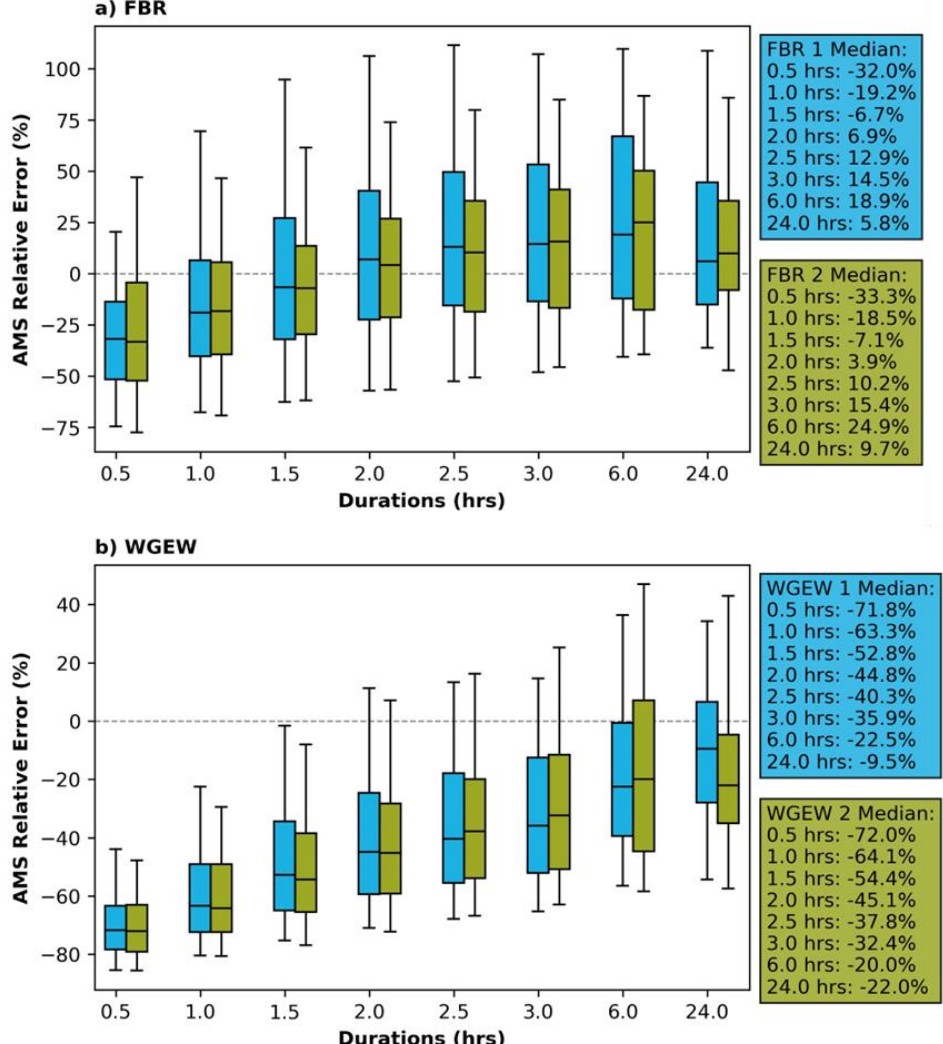

**Figure 10.** Percent relative error of precipitation AMS from IMERG compared with the gauge-based cells over FBR (**a**) and WGEW (**b**). The lower whiskers are located at the 5th percentile and the upper whiskers at the 95th percentile.

Figure 10 shows that the RE varies with duration and region. In FBR, the median of the RE ranges between −33% and 25%, whereas the lower REs are at 2 and 24 h. While, for WGEW, the median of the RE is between −72% and −9.5% where it tends to decrease with longer durations. It should be noted that the differences in the median RE between FBR and WGEW may partly come from the differences in the percentage of coverage. The gridded gauge-based precipitation over the WGEW has 8455 cells covering 93% of each IMERG grid. However, the FBR dataset has 2300 cells covering 100% of each IMERG grid. Nonetheless, the REs suggest that annual maximum precipitation intensities from gridded precipitation products (here IMERG at 0.1o resolution) can be very different from the precipitation measurements at rain station locations and the associated errors vary with both precipitation duration and intensity.

Figure 11 summarizes the RE of each cell IDF against the corresponding IMERG grid IDF for FBR 2 (Figure 11a–d) and WGEW (Figure 11e–h) regions for the 2, 5, 10, and 100 year return periods. Similar to Ombadi et al. [7] and Noor et al. [5], we noticed that the RE tends to decrease with the duration, but it increases with the return period. Even though the GEV and MLE showed the best goodness of fit in the KS-test, it is the most sensitive CDF to local changes and very sensitive to the return period. For instance, Figure 11d shows that the box sizes of the RE associated with the GEV and MLE are significantly larger than

the other CDFs. In contrast, both Gumbel distributions tend to be less sensitive to local changes than other distributions. Around 80% of the time, the IDFs fitted with Gumbel and MLE show a median RE range between ~20% and −40% in FBR 2 and between ~−15% and −70% in WGEW 1. Overall, Figure 11 shows that IMERG IDFs, fitted with Gumbel, better represent the averaged gauge-based IDFs for an area of $0.1° × 0.1°$. It must be noted that the choice of the empirical model can affect the results shown in this paper and it can be further investigated in future works.

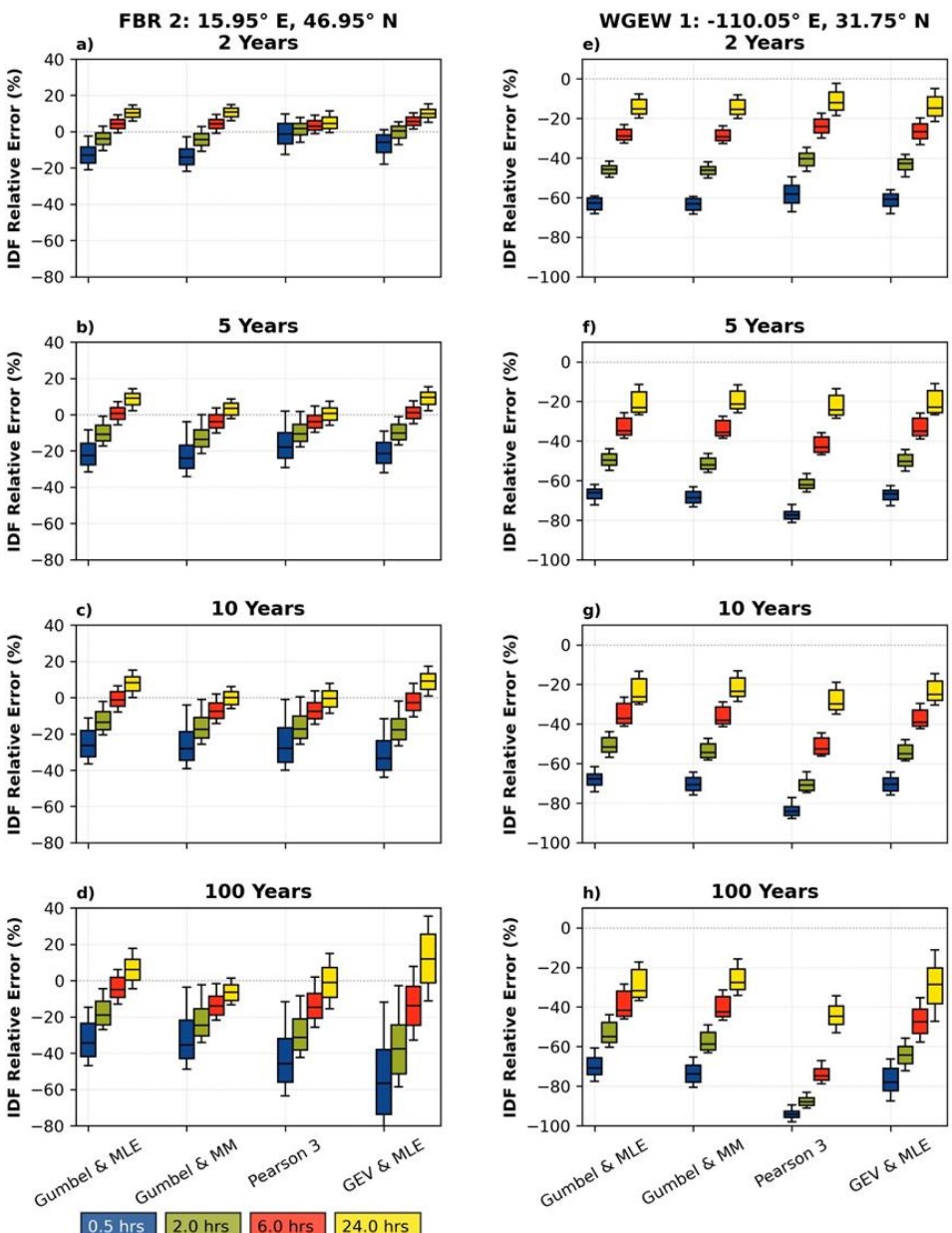

**Figure 11.** Percent relative error of precipitation AMS from IMERG compared to the gauge-based cells over FBR 2 (**left**) and WGEW 1 (**right**) for durations of 0.5, 2, 6, and 24 h. From top to bottom: for the 2, 5, 10, and 100 year return periods. The lower whiskers are located at the 10th percentile and the upper whiskers at the 90th percentile.

*3.6. IDF Relationships under Climate Change*

Martel et al. [33] reviewed 58 studies on rainfall extremes under climate change. They found that 100% of the investigated studies projected increases in rainfall extremes. Kourtis et al. [34] summarized more than 100 articles regarding the development and/or update of IDF curves considering climate change. Both literature reviews concluded that researchers have reported an increase in extreme rainfall intensities based on in situ observations for different regions around the globe. Researchers also noted that updating IDF curves must consider climate change projections, leading to non-stationary assumptions of rainfall extremes [33,34].

On another aspect, 73% of the studies summarized by kourtis et al. reported that Gumbel (24% of the examined studies) and GEV (49% of the examined studies) distributions are the main theoretical distributions used for the development of IDF curves considering the effects of climate change. Additionally, they pointed out that in many regions around the globe, especially in developing countries, adequate rainfall records are not available for various reasons, i.e., limitation in spatial coverage, short rainfall records, and low quality of data [34]. In this context and highlighting from our results that IMERG IDF curves calculated with a Gumbel distribution show better agreement with the gauge-based IDF curves, IMERG data could be used as baseline combined with technics to incorporate climate change. Some approaches to consider are Simple Constant Percentage Increase, Adaptive Percentage Increase, or Percentage Increases Based on the Clausius–Clapeyron Relationship. For more information regarding these approaches refer to Martel et al. [33].

To the authors' knowledge, no study has been conducted before to evaluate the performance of satellite-based IDF curves using sub-hourly data. Therefore, this study could ground further research for calibration or bias-correction of IDF curves based on Regional Climate Models (RCMs) and the most recent Convection-Permitting Climate Models (CPMs).

## 4. Conclusions

In this study, we evaluated the potential use of IMERG Final (V06) half hourly precipitation to derive IDF curves by comparing them against the derived IDF curves using data from two separate regions with high-density gauge networks. The FBR network is located in the southeastern Alpine forelands of Austria, while the WGEW network is in a semiarid region of the southeast of Tucson, Arizona. These two regions have very different types of land surface and climate regimes. Considering the high density of the FBR and WGEW gauge networks, the evaluations were performed at individual IMERG grids with a spatial resolution of $0.1° \times 0.1°$.

Before deriving the IDF curves, we showed that the performance of IMERG AMS varies with the study areas because IMERG estimations and the gauge measurements are affected by various errors such as limitations of satellite sensors, precipitation retrieval assumptions, loss of precipitation in the gauges and within the atmosphere column below clouds, incorrect identification of the type of surface, orography, precipitation regime, and number of gauges reported to the GPCC.

Additionally, we examined how the order of calculating AMS in gauges within an IMERG grid can affect the evaluation of the IMERG AMS. We found that IMERG AMS is more comparable to the gauge areal average before obtaining the AMS (AB-AMS). Alternatively, an areal average after obtaining the AMS (AA-AMS) can increase the amount of the maximum precipitation by ~18% compared to the AB-AMS approach.

We found that IMERG AMS and IDF curves have better performance over the FBR. However, through the qq-plots we determined that the performance over WGEW could be improved with the application of an adjustment factor to the IMERG AMS for each duration.

On the grid scale, IMERG IDF curves demonstrated a higher tendency to underestimate the precipitation intensity. However, in most cases, the biases converge to zero for durations longer than 5–9 h, depending on their return periods. Therefore, the KGEss and NSE implied that the use of Gumbel distributions in calculating the IDF curves from IMERG results in a better agreement with the gauge-based IDF curves.

With all the known errors in comparing satellite grid (representing areal average) to rain gauge (representing point measurement) precipitation, IMERG IDF curves are compared better with rain gauge IDF curves (in terms of RE) when the Gumbel distribution is used. For instance, the median RE at a duration of 6 h is almost zero for 5 and 10 year return periods over FBR 2.

Our results contribute to a better understanding of the potential use of IMERG to derive IDF curves. The use of IMERG (and similar products) can be an important alternative for calculating regional IDF curves in regions with sparse gauge networks. Acknowledging the limitations of the IMERG and the resulting IDF curves, we recommend a Gumbel distribution to calculate the IDF curves from IMERG, as it shows better agreement with our gauge analysis over the two regions studied here. Considering the current spatial resolution of IMERG, the IDF curves may not be as effective for areas smaller than ~0.1° × 0.1° latitude/longitude due to potential areal mismatch. Developing long-term homogenous precipitation products with improved spatiotemporal resolution and sub-monthly bias adjustment can improve the development of the IDF curves, which may need periodic revisions in a changing climate.

**Author Contributions:** Conceptualization, A.L. and A.B.; methodology, A.L.; software, A.L.; validation, A.L.; formal analysis, A.L.; investigation, A.L.; resources, A.L. and A.B.; data curation, A.L.; writing—original draft preparation, A.L.; writing—review and editing, A.B.; visualization, A.L.; supervision, A.B.; project administration, A.B.; funding acquisition, A.L. and A.B. All authors have read and agreed to the published version of the manuscript.

**Funding:** A.L. was funded by the Fulbright-SENACYT scholarship.

**Data Availability Statement:** WGEW datasets were provided by the USDA-ARS Southwest Watershed Research Center. Funding for these datasets was provided by the United States Department of Agriculture, Agricultural Research Service. The WegenerNet is funded by the Austrian Ministry for Science and Research, the University of Graz, the state of Styria, and the city of Graz. Most of the data is available to the public. IMERG Final L3 Half Hourly (V06) product can be accessed from https://disc.gsfc.nasa.gov/ (accessed on 30 November 2021). WegenerNET FBR 200 × 200 m gridded half-hourly precipitation can be accessed at https://wegenernet.org/portal/ (accessed on 30 November 2021). Data related to the WGEW can be accessed at https://www.tucson.ars.ag.gov/dap/ (accessed on 20 October 2021).

**Acknowledgments:** The authors gratefully acknowledge the support and assistance of Carl Unkrich (USDA-ARS) and Juergen Fuchsberger (Wegener Center).

**Conflicts of Interest:** The authors declare no conflict of interest.

## Appendix A

This section presents the figures for the other two studied grids: FBR 1 and WGEW 2. The section is organized following the same order from Section 3. Results and Discussion. Figures A1–A3 corresponds to the subsection Statistical comparison between the IMERG AMS and the gauge-based AMS. Figures A4 and A5 look into the performance of the IMERG IDF equations. Additionally, Figures A6–A9 show all the derived sets of IDFs with each of the CDF considered in the study; where Figures A10 and A11 are the plots for its corresponding biases. Lastly, Figure A12 relates to the Assessment of IMERG AMSs and IDF equations at local scale.

*Appendix A.1. Statistical Comparison between the IMERG AMS and the Gauge-Based AMS*

**FBR 1: 15.85° E, 46.95° N**

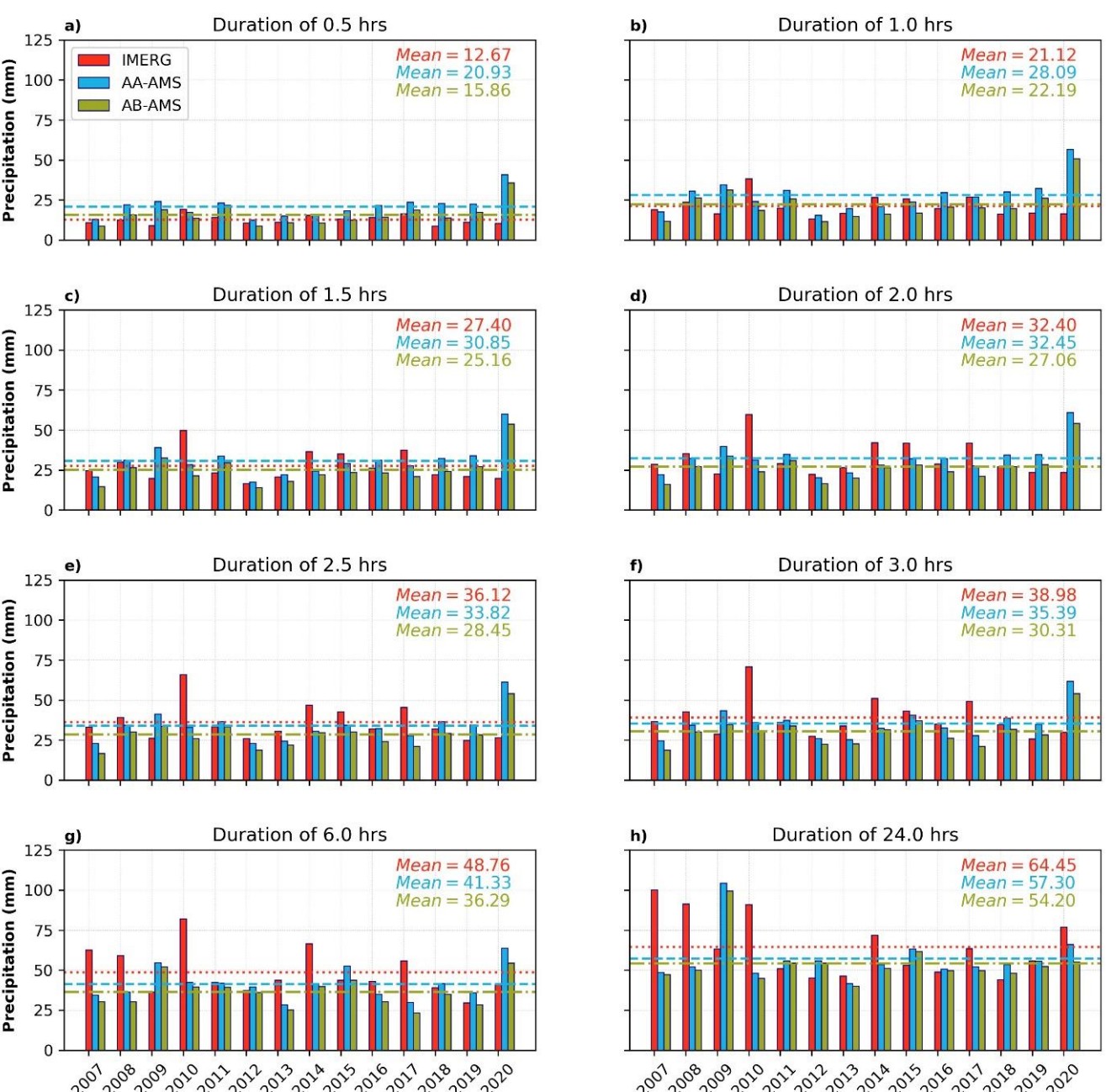

**Figure A1.** Annual maximum time series for FBR 1 by durations. The subfigures illustrate event durations of 0.5-h (**a**), 1-h (**b**), 1.5-h (**c**), 2-h (**d**), 2.5-h (**e**), 3-h (**f**), 6-h (**g**), 24-h (**h**). In all the subfigures, thered bars are the IMERG AMS, the light blue bars are the AA-AMS, and the green bars are the AB-AMS. The dashed lines are the mean AMS for each case.

**WGEW 2: -109.95° E, 31.75° N**

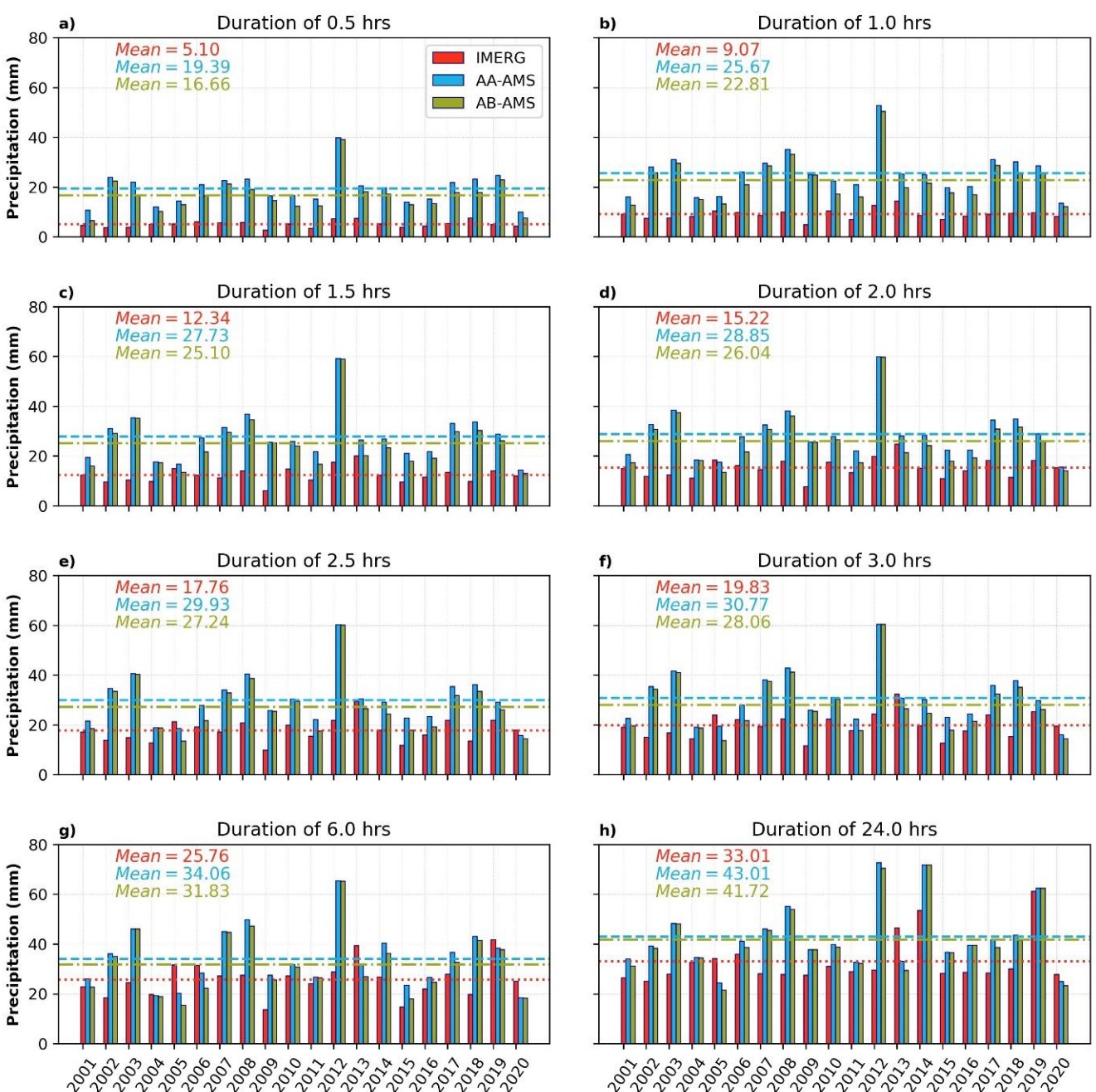

**Figure A2.** Annual maximum time series for WGEW 2 by durations. The subfigures illustrate event durations of 0.5-h (**a**), 1-h (**b**), 1.5-h (**c**), 2-h (**d**), 2.5-h (**e**), 3-h (**f**), 6-h (**g**), 24-h (**h**). In all the subfigures, thered bars are the IMERG AMS, the light blue bars are the AA-AMS, and the green bars are the AB-AMS. The dashed lines are the mean AMS for each case.

*Appendix A.2. AA-AMS versus AB-AMS*

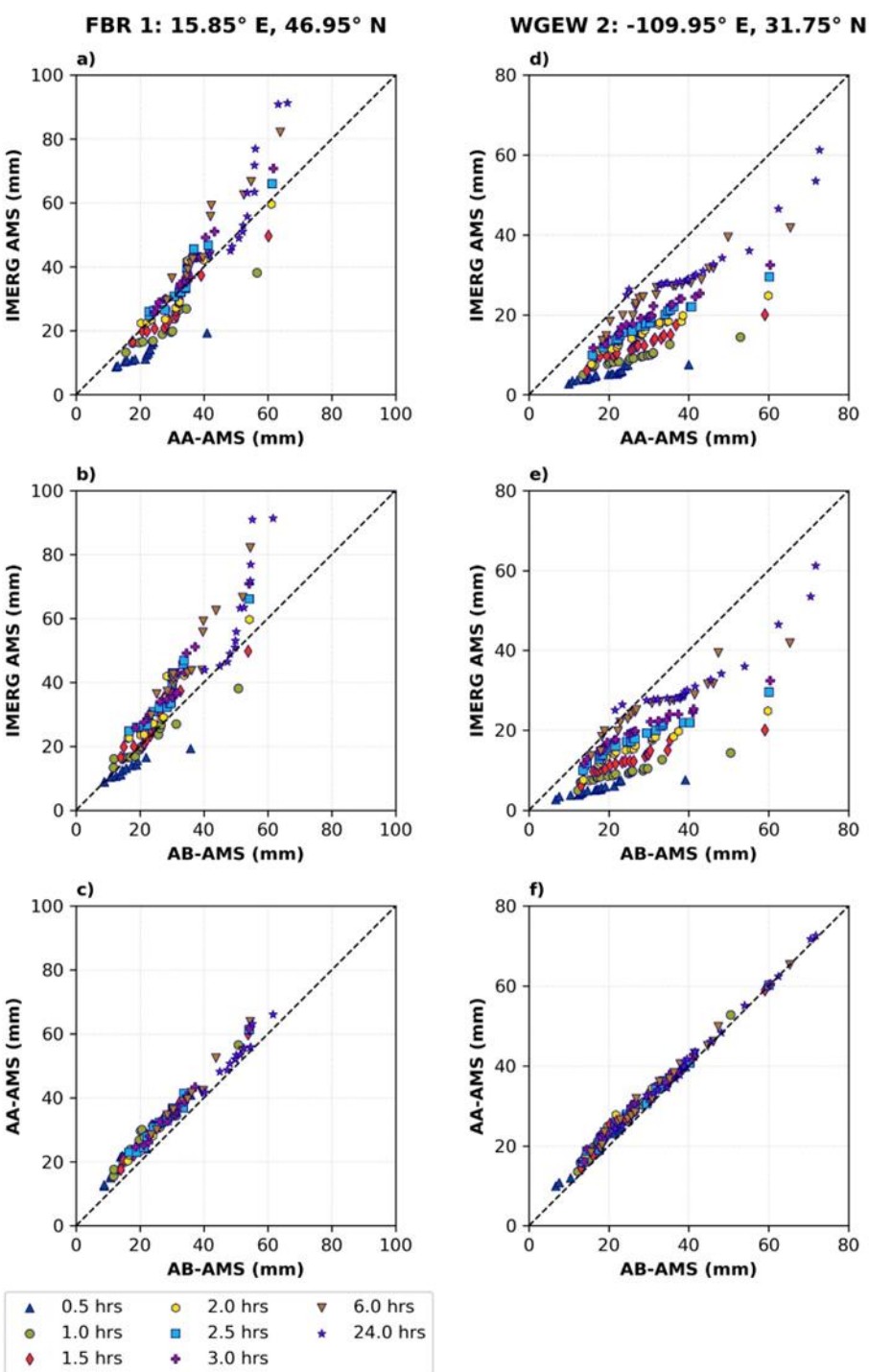

**Figure A3.** QQ-plots comparing the AMSs for FBR 1 (**left**) and WGEW 2 (**right**). From top to bottom: IMERG AMS vs. AA-AMS, IMERG AMS vs. AB-AMS, and AA-AMS vs. AB-AMS. The markers are the durations 0.5, 1, 1.5, 2, 2.5, 3, 6, and 24 h. The 45° dashed line is a reference line.

*Appendix A.3. Grid-Scale: Performance of the IMERG IDF Equations*

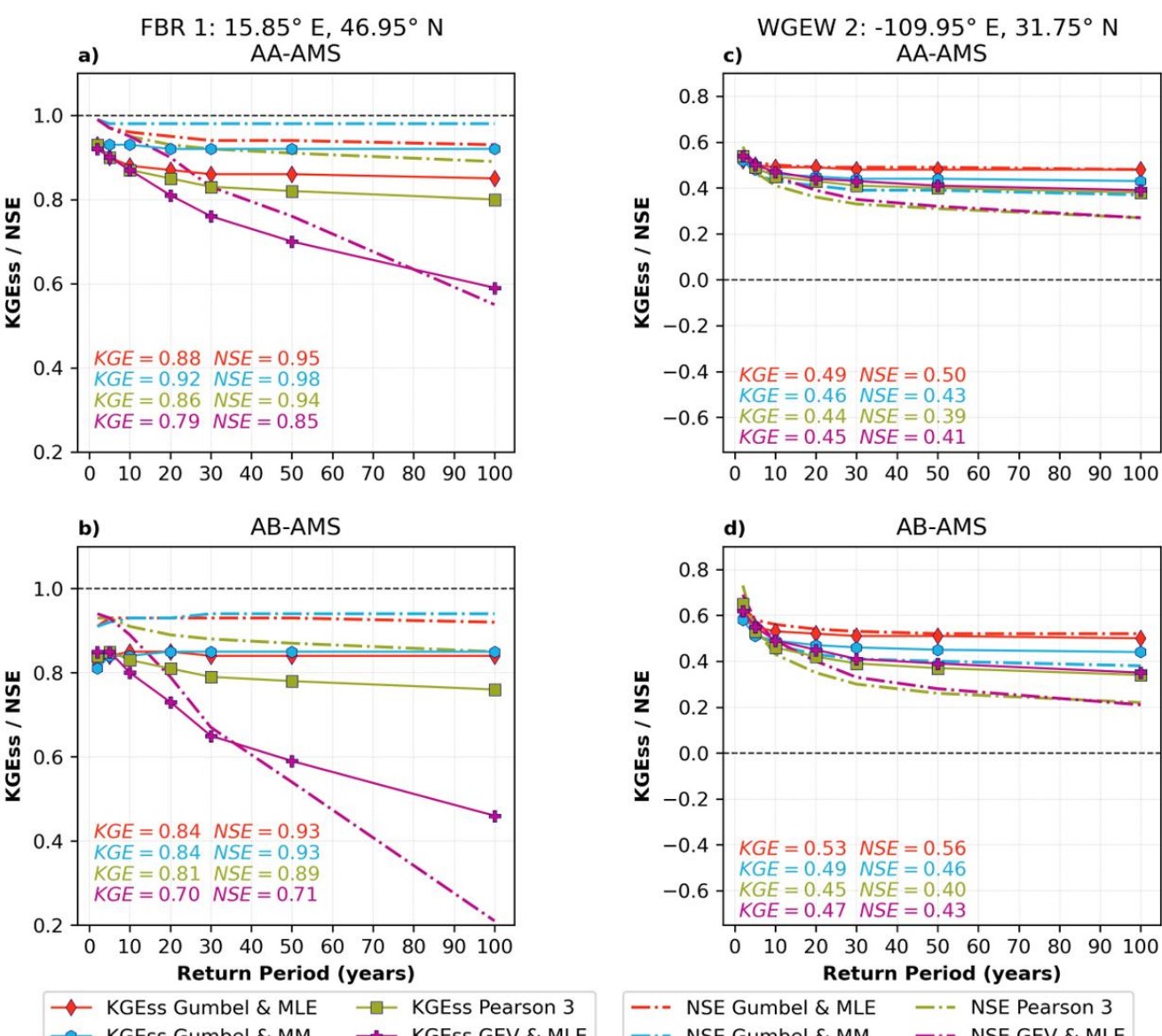

**Figure A4.** KGEss and NSE obtained for the four CDFs and the return periods 2, 5, 10, 20, 30, 50, and 100 years over FBR 1 (**left**) and WGEW 2 (**right**). IMERG IDF vs. AA-AMS IDF (subplots **a,c**) and IMERG IDF vs. AB-AMS IDF (subplots **b,d**). The solid lines with markers are for the KGEss and the dashed lines are for the NSE.

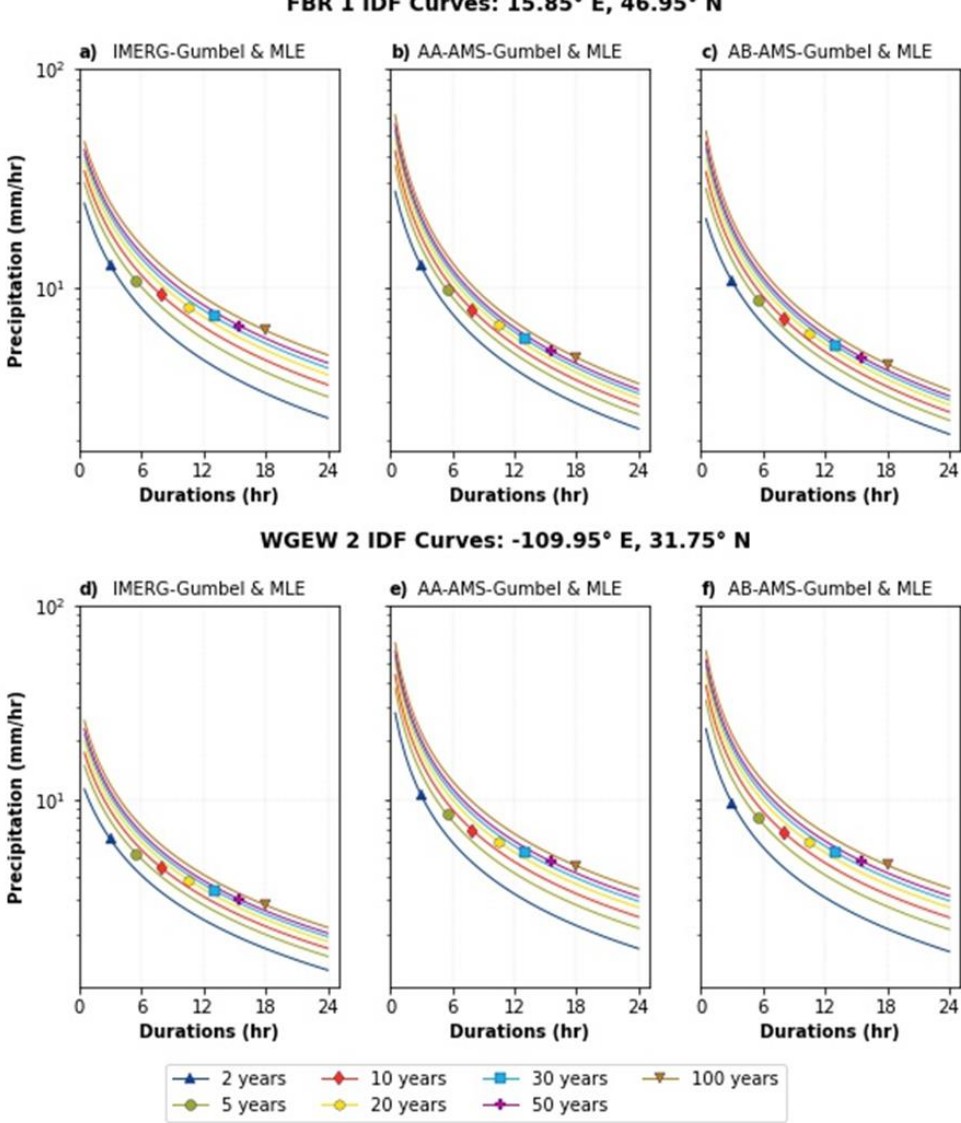

**Figure A5.** IDF curves fitted with Gumbel and MLE over FBR 1 (**top**) and WGEW 2 (**bottom**) for the 2, 5, 10, 20, 30, 50, and 100 year return periods and durations from half-hour to 24 h. From left to right: IMERG, AA-AMS, and AB-AMS. The marker's purpose is only to distinguish the IDF curves.

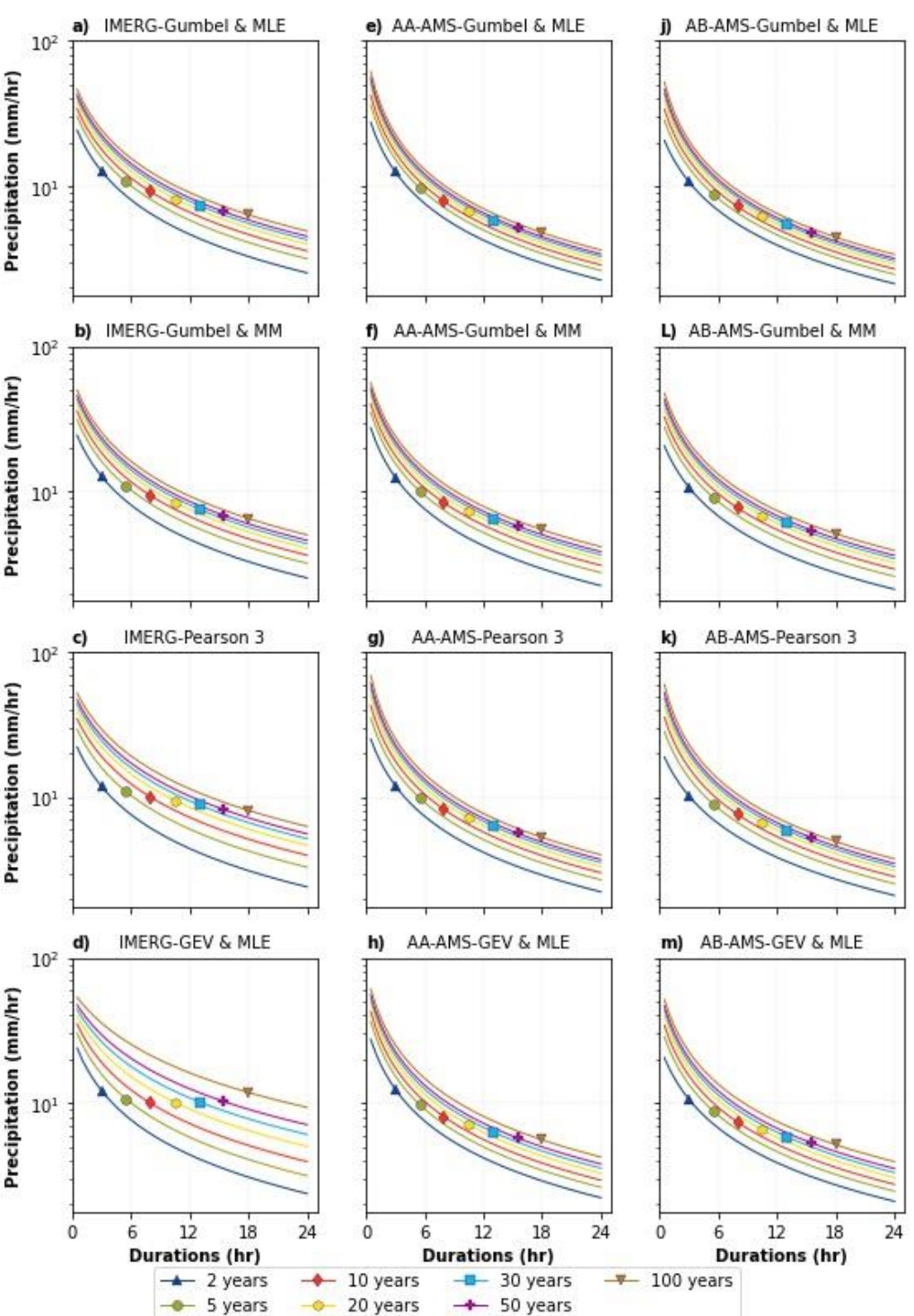

**Figure A6.** IDF curves fitted with the four CDFs over FBR 1 for the 2, 5, 10, 20, 30, 50, and 100 year return periods and durations from half-hour to 24 h. From left to right: IMERG, AA-AMS, and AB-AMS. From top to bottom: Gumbel and MLE, Gumbel and MM, Pearson 3, and GEV and MLE The marker's purpose is only to distinguish the IDF curves.

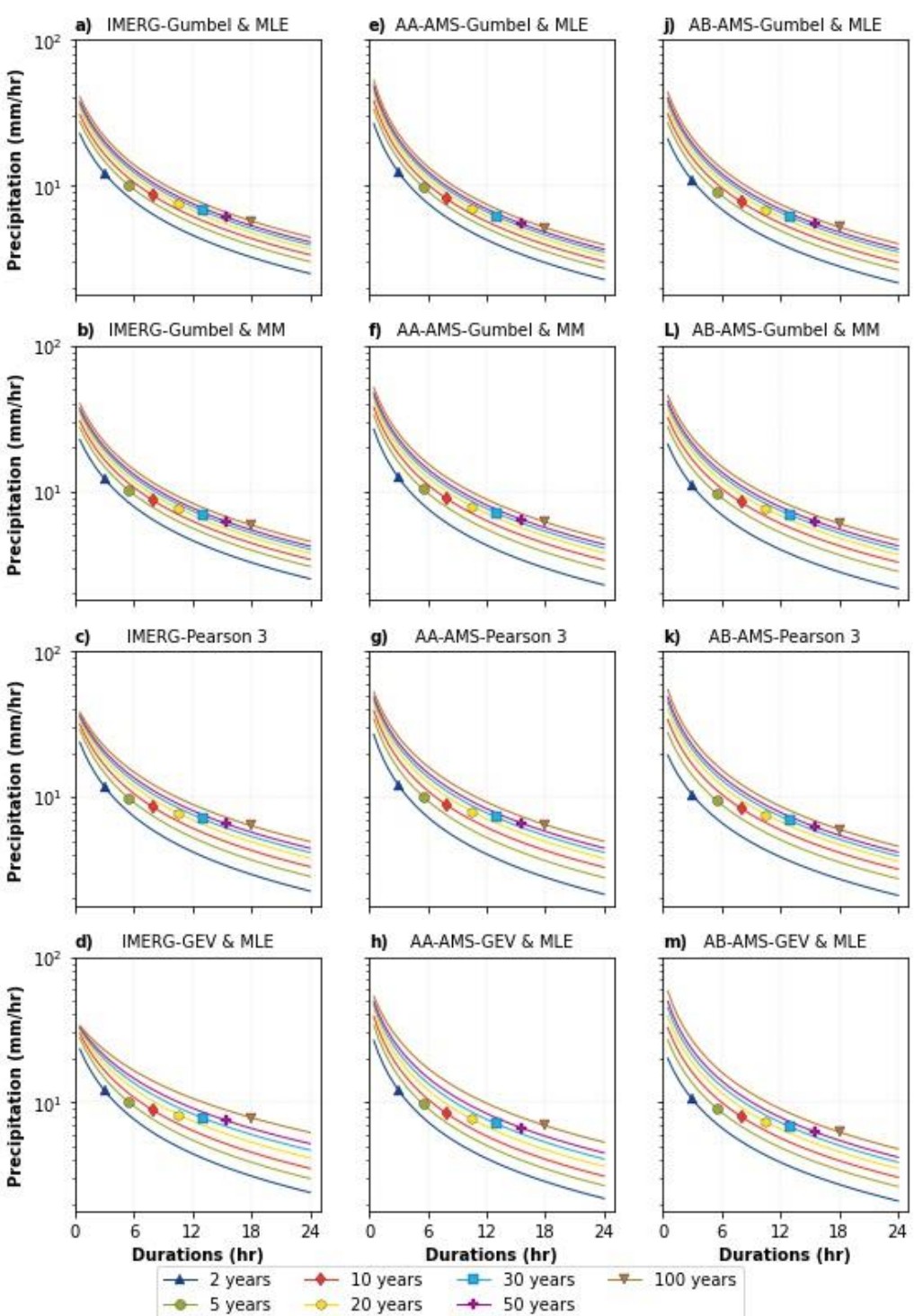

**Figure A7.** IDF curves fitted with the four CDFs over FBR 2 for the 2, 5, 10, 20, 30, 50, and 100 year return periods and durations from half-hour to 24 h. From **left** to **right**: IMERG, AA-AMS, and AB-AMS. From **top** to **bottom**: Gumbel and MLE, Gumbel and MM, Pearson 3, and GEV and MLE The marker's purpose is only to distinguish the IDF curves.

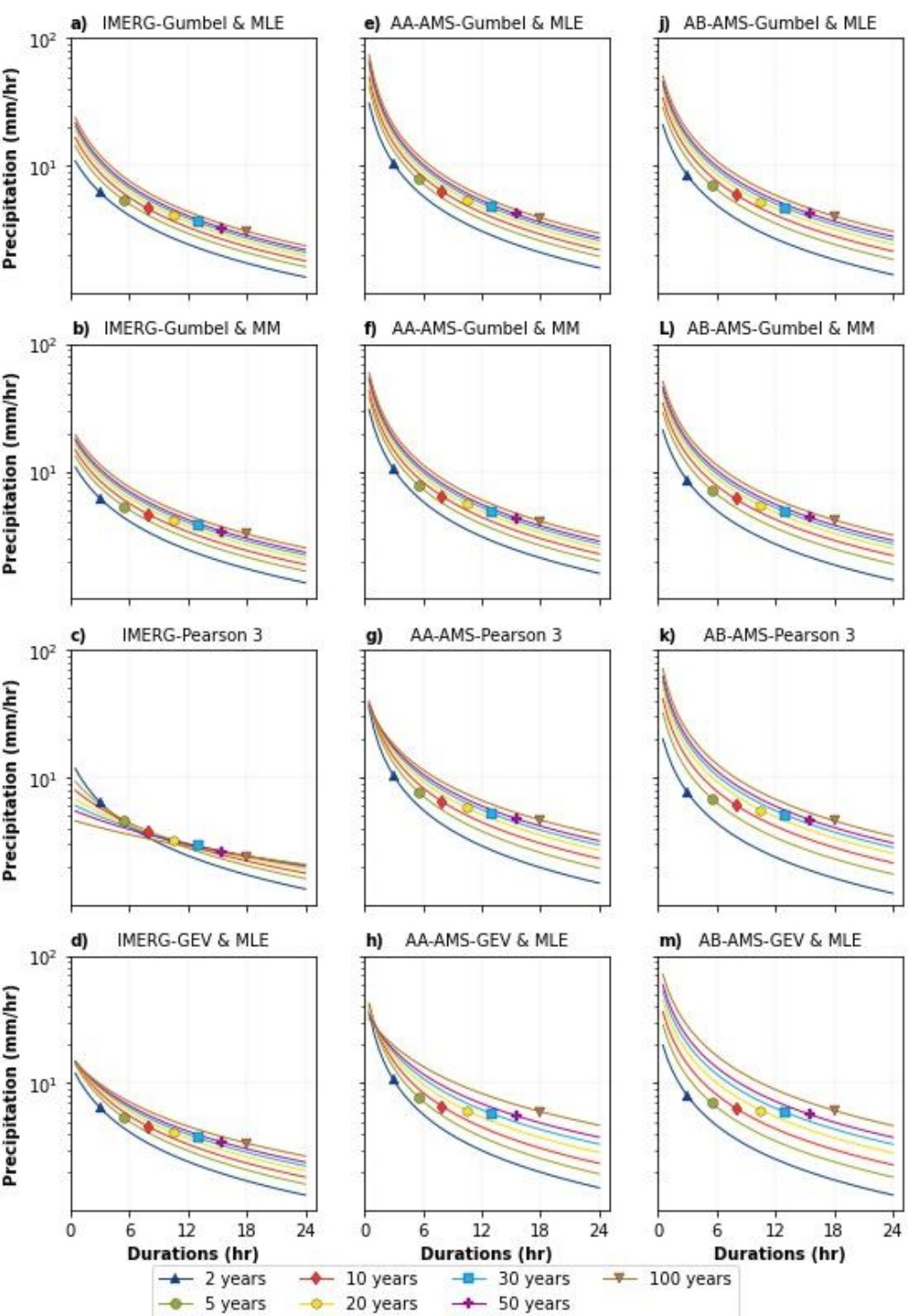

**Figure A8.** IDF curves fitted with the four CDFs over WGEW 1 for the 2, 5, 10, 20, 30, 50, and 100 year return periods and durations from half-hour to 24 h. From **left** to **right**: IMERG, AA-AMS, and AB-AMS. From **top** to **bottom**: Gumbel and MLE, Gumbel and MM, Pearson 3, and GEV and MLE The marker's purpose is only to distinguish the IDF curves.

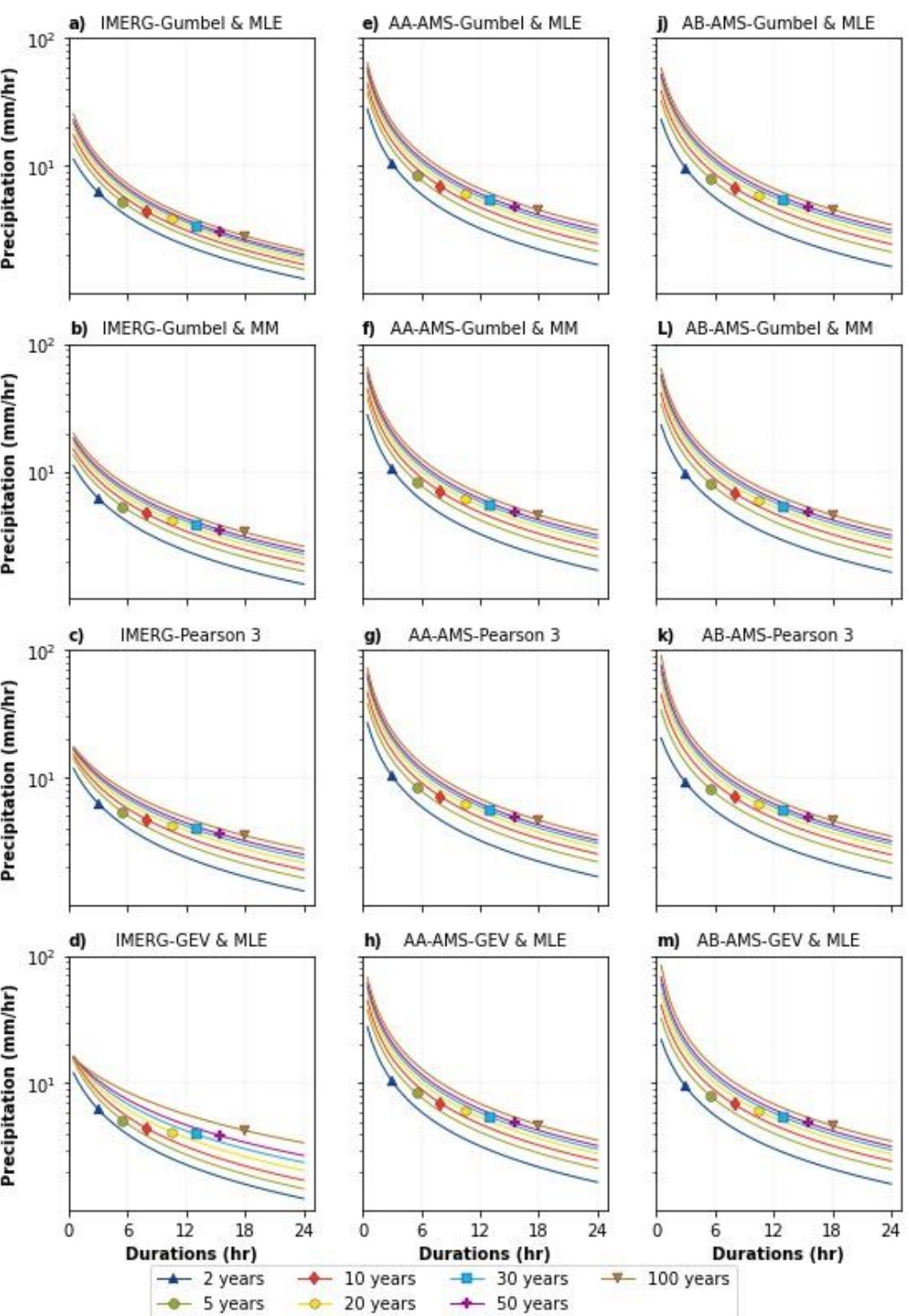

**Figure A9.** IDF curves fitted with the four CDFs over WGEW 2 for the 2, 5, 10, 20, 30, 50, and 100 year return periods and durations from half-hour to 24 h. From **left** to **right**: IMERG, AA-AMS, and AB-AMS. From **top** to **bottom**: Gumbel and MLE, Gumbel and MM, Pearson 3, and GEV and MLE The marker's purpose is only to distinguish the IDF curves.

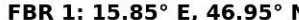

**Figure A10.** Precipitation intensity biases between IMERG IDFs and the gauge-based IDFs for 2, 5, 10, and 100 year return periods and event durations between 0.5 and 12 h over the FBR 1. IMERG IDF vs. AA-AMS IDF (**left**) and IMERG IDF vs. AB-AMS IDF (**right**).

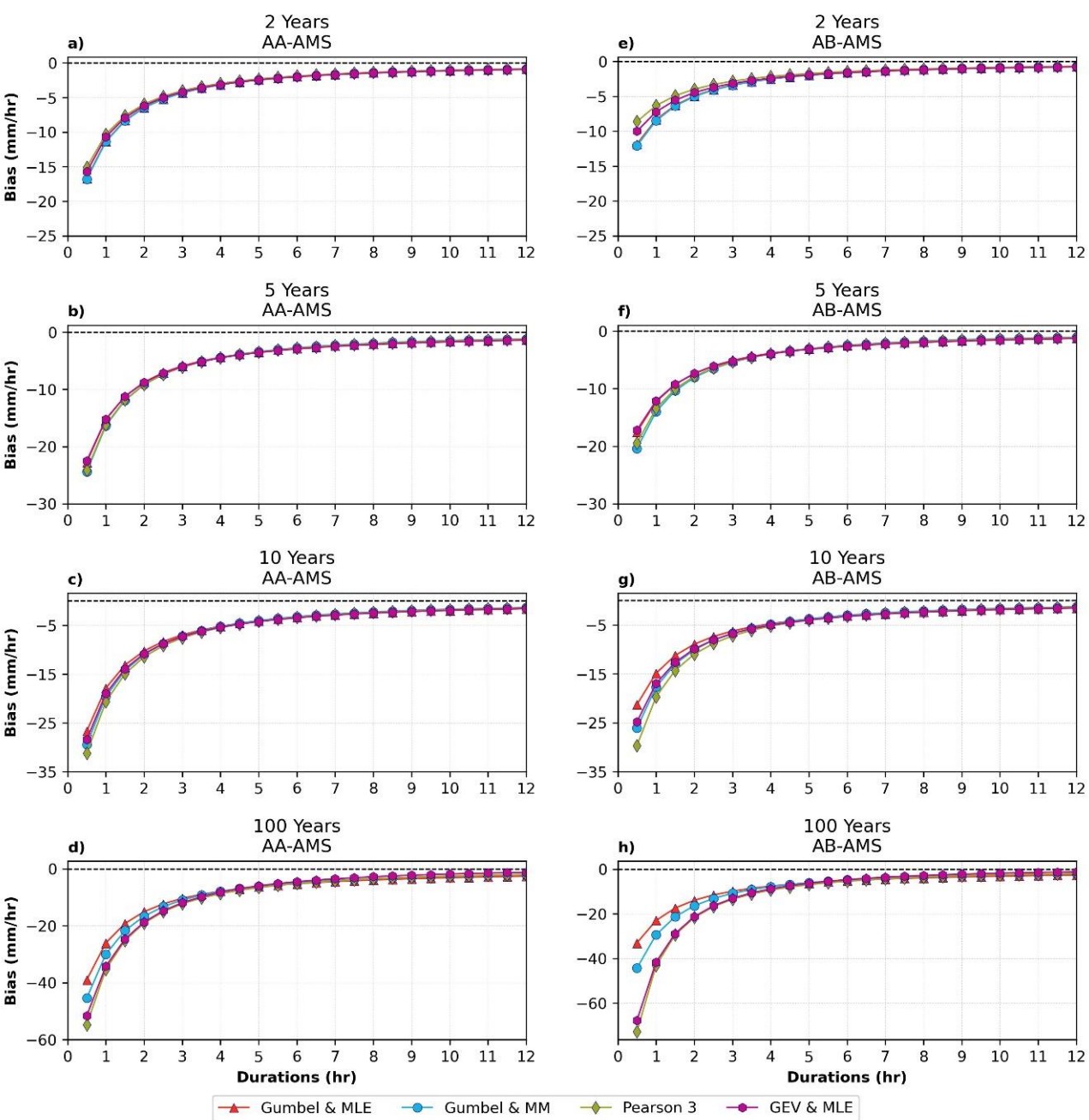

**Figure A11.** Precipitation intensity biases between IMERG IDFs and the gauge-based IDFs for 2, 5, 10, and 100 year return periods and event durations between 0.5 and 12 h over the WGEW 2. IMERG IDF vs. AA-AMS IDF (**left**) and IMERG IDF vs. AB-AMS IDF (**right**).

*Appendix A.4. Assessment of IMERG AMSs and IDF Equations at Local Scale*

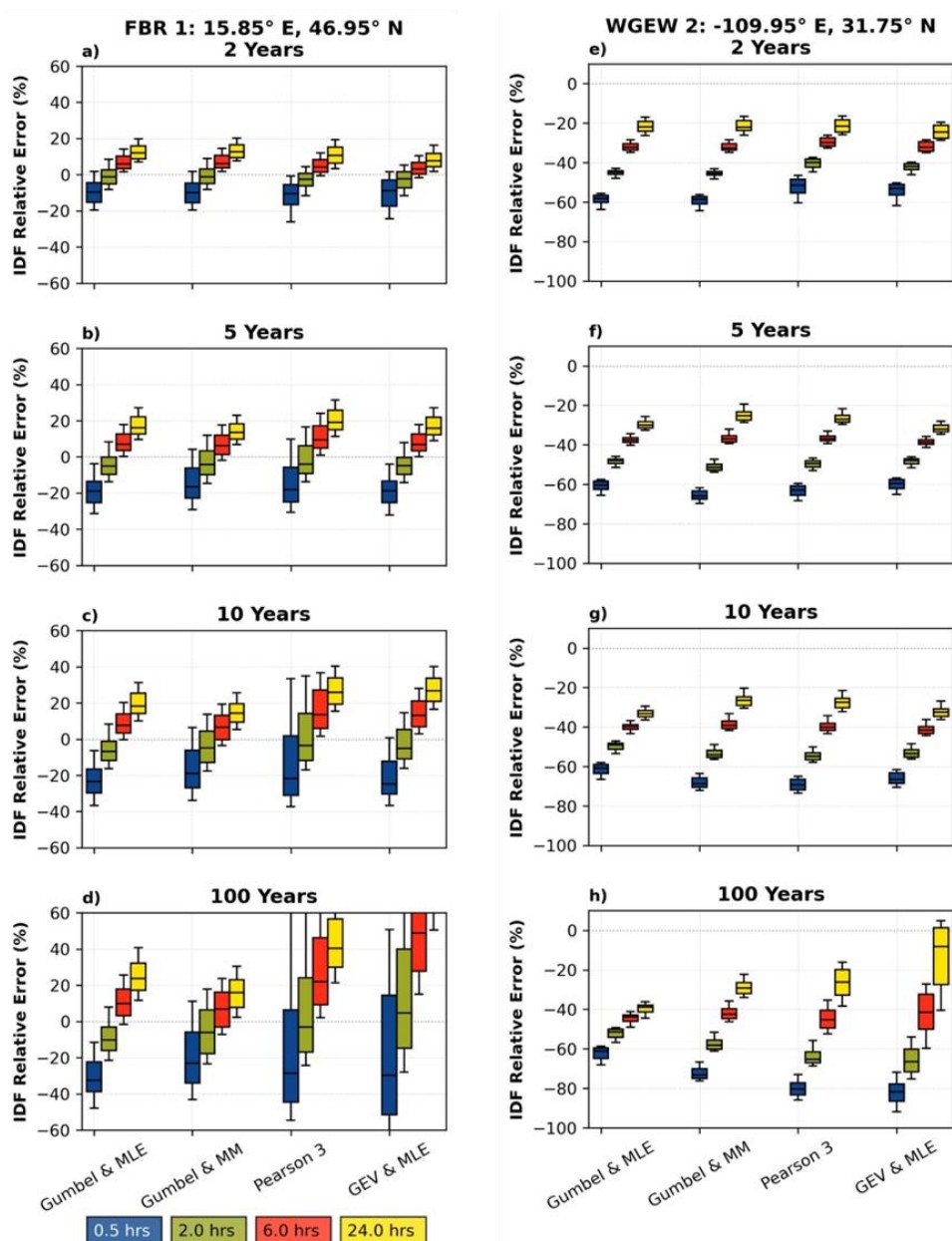

**Figure A12.** Percent relative error of precipitation AMS from IMERG compared to the gauge-based cells over FBR 1 (**left**) and WGEW 2 (**right**) for durations of 0.5, 2, 6 and 24 h. From top to bottom: for the 2, 5, 10, and 100 year return periods. The lower whiskers are located at the 10th percentile and the upper whiskers at the 90th percentile.

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
