# Peer review of "Understanding Intensity–Duration–Frequency (IDF) Curves Using IMERG Sub-Hourly Precipitation against Dense Gauge Networks"

_remotesensing, doi:10.3390/rs14195032_

Round 1

Reviewer 1 Report

The manuscript entitled "Understanding Intensity-Duration-Frequency (IDF) curves using IMERG sub-hourly precipitation against dense gauge networks" presents a study of IDF curves derived from the high-resolution remote sensing product dataset IMERG for two regions, Austria and the United States. In general, the authors show an application of the statistical and development of this needed tool for hydrological engineering, the IDF curves.

Before considering this paper suitable for publication in the Remote Sensing journal, the authors must clarify two main concerns about the paper. Firstly, the article structure must be fixed, considering a Conclusion section. One suggestion is to change Results to Results and Discussion and the Discussion as Conclusion since it presents all work done.

Secondly, and most importantly, the empirical formula of Talbot used in the study, as shown in Equation 4, must be compared and validated before using it for both regions. Why do not use another empirical formula, such as Kimijima or Bermard? The authors must justify using Talbot, not just say it is widely used.

Minor points reviews:

- Reference section is not complete.

- Not followed the author's guide format.

Reviewer 2 Report

This paper deals with the potential use of IMERG Final half hourly precipitation to derive IDF curves. 

The topic is interesting and it meets the aims and scope of "remote sensing".

In general the paper is well-written. The methods are not new but they have been used competently by the authors. The results are clear and show some degree of novelty. 

I can't find any particular flaws in this work. However,  it may be improved, highlighting better what is the real advance of this work for the related literature. Basically, the discussion section is missing. What is called "discussion" are general conclusions in my view. The authors should better stress the novelty of their research in the light of recent papers in the literature. I reported some papers at the end of this report [1-3] that can be useful to discuss the results obtained by the authors and to highlight the knowledge conveyed through this work.

Another point of the discussion should be related to the effect of climate change on the IDF curve. I can't find any reference to papers linked to this essential topic. I suggest considering for example [4-6] reported at the end of my review. 

I believe that the authors can easily address these two points, organizing a suitable discussion section for this paper integrating the suggested references and , possibly, similar ones. For this reason, my recommendation is "Minor Revision".

Cited works

[1] DOI: 10.5194/hess-21-2389-2017

[2] DOI: 10.3390/rs11050555

[3] DOI: 10.3390/w14111699

[4] DOI:10.2166/ws.2022.152

[5] DOI: 10.1016/j.jhydrol.2021.126756

[6] DOI: 10.1061/(ASCE)HE.1943-5584.0002122

Reviewer 3 Report

This is a well-motivated work, with a focus of deriving Intensity-Duration-Frequency (IDF) curves from the Global Precipitation Mission's (GPM) observations. IDF curves are a key input in hydrological models. The authors utilize the final Integrated Multi-satellitE Retrievals for GPM (IMERG) sub-hourly products. One justification which would be useful for readers, is to why they chose this product over the Early and Late IMERG products. Overall, the data collection, analysis and interpretation of the results are satisfactory, so if the authors provide details on this, I would recommend the article for publication. 

Round 2

Reviewer 1 Report

The authors answered all of my questions very clearly. Considering the manuscript suitable for publication.